# Smooth Real-time Rendering via Implicit Nested Neighborhoods

## Abstract

*Implicit neural representations* (INRs) for surfaces have been mostly used as intermediary representations before triangle mesh extraction. Extracting meshes is not a real-time task and introduces unnecessary discretization to rendering, making it difficult to fully use the smoothness of INRs in applications. Smooth INRs are broadly used for approximating surface *signed distance functions* (SDFs) through an implicit regularization (Eikonal equation) using their available high-order derivatives. Such property also makes it easier to integrate those INRs in pipelines that explore differentiable properties of the underlying surface. The current real-time state-of-the-art approach uses grid-based data-structures that introduce discretization, resulting in a non-smooth representation.

We propose an end-to-end smooth ($C^\infty$) INR framework to represent and render surfaces in real-time using neural SDFs endowed with smooth attributes such as normals and textures. Our approach leverages from a novel localized SDF training based on nested neighborhoods, a multiscale surface representation, and residual training. The framework does not depend on spatial data-structures, nor surface extraction. We show that our representation renders detailed smooth surfaces in real-time while the previous works can only render coarse non-smooth surfaces. We also present applications of our representation, including integration with a pipeline for dynamic surfaces and a way to improve performance of surface extraction via marching cubes.

## 1 Introduction

Real-time rendering enables interactive applications on 3D scenes. One important choice before rendering is how to represent the scene objects. In this setting, *implicit neural representations* (INRs) are emerging as a compelling option, which encodes the surface as the zero-level set of a *neural network*. They are memory efficient, continuous, infinitely differentiable when periodic activations are used, scalable, and naturally adapted to machine learning pipelines.

Even though those properties are present, INRs for surfaces have been mostly used as intermediary representations in neural pipelines, which usually output triangle meshes. Although that representation excels at localized tasks because it is explicit, it is not memory efficient, nor continuous, nor scalable. One of the reasons behind the use of triangle meshes is the non-triviality of rendering INRs for surfaces in real-time while maintaining the aforementioned properties. The current approach for real-time rendering of INRs resorts to discrete spatial data-structures which, analogously to triangle meshes, cannot maintain smoothness. Easy access to derivatives makes a model ready for integration with differentiable pipelines from the inception, increasing its possibilities for applications.

We propose a real-time rendering framework using INRs which maintain surface and attribute smoothness. Our approach trains *residual SDFs* on neighborhoods of the zero-level set, resulting in a *multiscale* representation. Surface attributes (normals and texture) are also defined and trained in the surface neighborhood. Rendering-wise, we propose a *multiscale sphere tracing* and a normal computation based on general matrix multiply (GEMM) (Dongarra et al., 1990) to render the surfaces.

Summarizing, our contributions are: **(1)** Smooth multiscale INR for surface representation; **(2)** Efficient neighborhood training for SDF (using a residual scheme), normals, and texture; **(3)** Multiscale sphere tracing; **(4)** GEMM-based normal computation. **(5)** Applications in differentiable pipelines (dynamic surfaces) and in an adaptive sampling for fast mesh extraction via marching cubes.

## 2 RELATED WORK

Implicit representations are an essential topic in computer graphics (Velho et al., 2007). SDFs are important examples of such functions (Bloomenthal & Wyvill, 1990) and arise from solving the Eikonal problem (Sethian & Vladimirsky, 2000). Recently, *multilayer perceptrons* (MLPs) have been used to model SDFs (Park et al., 2019; Gropp et al., 2020; Novello et al., 2022). *Sinusoidal networks* (SIRENs) Sitzmann et al. (2020) are an example of such, being MLPs using sinusoidal activation.

*Marching cubes* (Lorensen & Cline, 1987) and *sphere tracing* (ST) (Hart, 1996) are classical visualization methods for rendering level sets of SDFs. Neural versions of those algorithms were proposed by (Liao et al., 2018; Chen & Zhang, 2021; Liu et al., 2020). While the initial works in neural SDFs use marching cubes to visualize the resulting level sets, recent performance-driven approaches have been using ST, since no intermediary representation is needed for rendering (Davies et al., 2020; Takikawa et al., 2021). Our proposed multiscale ST considers a similar path.

**Surface representations and rendering:** Recent works propose INRs for disentangling base geometry and detail. Wang et al. (2022) describe an INR using base and displacement networks to compute a detailed triangle mesh extracted via marching cubes for rendering. This approach is similar to our residual SDF learning, however it relies on function composition instead of addition like ours. A consequence is that the inference of the base and displacement network must be sequential, different from our approach which may be parallelized. Another difference is that our residual training is done only at the neighborhood of the zero-level set, which improves the network ability to represent the function since it is restricted to a small neighborhood instead of the entire domain. Morreale et al. (2022) employ INRs to model surfaces parametrically using parameterizations. Differently from our approach, they do not deal with the rendering problem. Sharp & Jacobson (2022) describe a way to perform geometric queries for neural SDFs using range analysis. (Genova et al., 2020) uses multiple small implicit objects to increase detail of the representation. None of those approaches support textures. Contextualization with triangle meshes is in Section A.1.

**Real-time neural SDFs:** Fast inference is needed to sphere trace SDFs in real-time. Davies et al. (2020) show that this is possible using *general matrix multiply* (GEMM) (Dongarra et al., 1990; Müller, 2021), but the capacity of their networks can not represent geometric detail. Other works in neural SDFs store features in the nodes of *octrees* (Takikawa et al., 2021; Martel et al., 2021), or limit the frequency band in training as in BACON (Lindell et al., 2021). However, octree-based approaches reintroduce discretization to the pipeline. NGLOD (Takikawa et al., 2021) is the SOTA real-time method for rendering neural SDFs. It uses a *sparse voxel octree* (SVO) to represent the neural SDF and render its level set using a *sparse* ST algorithm. The vertices of the voxels store features. Then, for a point $p$ and a level $L$ of the SVO, the features are interpolated inside each voxel containing $p$ up to the level $L$. The resulting interpolated points are summed and passed to a MLP $f_L$. Thus, besides the SVO structure, NGLOD uses a sequence of $L$ MLPs to represent the LoD. Moreover, the interpolation implies in INRs with non-continuous gradients at the voxels boundaries leading to artifacts (Sec. 4.1). Ours supports smooth normals, by leveraging sinusoidal MLPs to fit each level of the SDFs using (Novello et al., 2022). Finally, NGLOD does not support textures as our method does.

**Attribute mapping:** *Normal mapping* (Cohen et al., 1998; Cignoni et al., 1998) is a classic method to transfer detailed normals between meshes. Besides depending on interpolation, normal mapping also suffers distortions of the parameterization between meshes, which are assumed to have the same topology. Recently, Wang et al. (2022) introduced detail transfer (normals) in the context of INRs. It is based on features computed from a point cloud encoder and a convolutional module to propagate sparse on-surface point features to the off-surface area. Queried features are obtained using bilinear interpolation. Our approach is simpler. Inspired by (Bertalmío et al., 2001), we use a regularization to make attributes constant along the normals near the zero-level set. That maintains smooth attributes, without the need of any interpolation or parametrization.

*Texture mapping* (Catmull, 1974) is a technique for cost-effective rendering that maps images to surfaces using parametrizations. In neural rendering, *texture fields* (Oechsle et al., 2019) shares similarities with our neural attribute mapping but approaches a different problem. Ours processes a point cloud with colors, while texture fields demand a 3D shape and input images, using view dependent depth maps. We use the surface's neighborhood to define color along normals. Texture fields is not real-time due to its use of 4-6 ResNet blocks and complex networks for latent code generation. In contrast, ours adopts small MLPs for efficient representation. GET3D (Gao et al., 2022) uses texture fields for the textures in its 3D model generation, sharing an analogous contextualization.

# 3 Nested Neighborhoods of neural SDFs

## 3.1 Overview

Given the iterative nature of Sphere Tracing (ST), a way to increase its performance is to optimize or avoid iterations. We propose to use small neural SDFs to approximate earlier iterations and mapping the normals and the texture of the desired neural SDF, avoiding later iterations. Both tasks can be accomplished by mapping neural SDFs using nested neighborhoods, without introducing any additional discretization. Fig. 1 shows our pipeline for smooth real-time rendering of 3D objects.

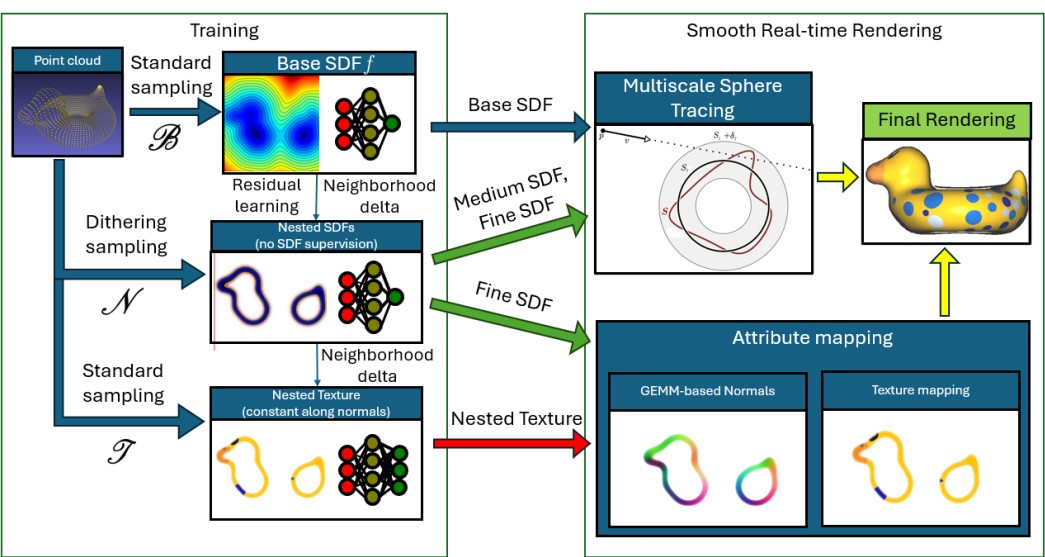

Figure 1: Our end-to-end INR framework for smooth real-time rendering. Starting from an oriented point cloud with colors, we combine sampling techniques and loss regularizations ($\mathcal{B}$ and $\mathcal{N}$) to create a base, a medium, and a fine SDF to implicitly represent the surface in multiscale. The base SDF is defined for the entire domain, while the others are residuals, defined in (nested) neighborhoods of the surface. The colors are also trained in a neighborhood, regularized (by $\mathcal{T}$) to be constant along normals. The resulting multiscale representation can be rendered using novel sphere tracing and attribute mapping algorithms.

The basic idea comes from the following fact: if the zero-level set of a neural SDF $f$ is contained in a neighborhood $V$ of the zero-level set of another neural SDF, then we can map $f$ into $V$. We follow the notation in Fig. 2 to present an overview of our method. Let $S_1, S_2, S_3$ be surfaces pairwise close with SDFs $f_1, f_2, f_3$ sorted by complexity. We use $S_1$ and $S_2$ to illustrate the multiscale ST and $S_3$ to illustrate the attribute mapping.

**Multiscale ST:** Suppose that the ray $p_0 + tv$, with origin at a point $p_0$ and direction $v$, intersects $S_2$. To compute its $S_2$ point $q_2$, we first use $f_1$ to sphere trace the boundary of a neighborhood of $S_1$ (gray) containing $S_2$. This results in $q_1$. Then we continue sphere tracing $S_2$ using $f_2$, reaching $q_2$. In other words, we are mapping the values of $f_2$ to the neighborhood of $S_1$.

**Neural Attribute Mapping:** For shading, we need a normal at $q_2$, which is given by $N_2 = \nabla f_2(q_2)$. Instead, we propose to pull the finer details of $S_3$ to $S_2$ to increase fidelity. This is done by mapping the normals from $S_3$ to $S_2$ using $N_3 = \nabla f_3(q_2)$. To justify this choice, note that $q_2$ belongs to a neighborhood of $S_3$. Thus, $N_3$ is the normal of $S_3$ at its closest point $q_3 = q_2 - \epsilon N_3$, where $\epsilon$ is the distance $f_3(q_2)$ from $q_2$ to $S_3$. This transfers the normal $N_3$ to $q_2$. Observe that $N_3$ is also the normal of the $\epsilon$-level set of $f_3$ at $q_2$ (red dotted). Similarly, the texture color is mapped from $q_3$ to $q_2$ by making it constant along $q_3 + tN_3$ in the neighborhood.

## 3.2 DEFINITIONS

A *neural SDF* $f : \mathbb{R}^3 \to \mathbb{R}$ is a smooth neural network approximating the Eikonal eq., i.e. $\|\nabla f\| \approx 1$. This work deals with the problem of rendering the *zero-level set* $f^{-1}(0)$ using a sphere tracing approach. Thus, given a point $p_0$ and a direction $v$, we must iterate $p_{i+1} = p_i + vf(p_i)$. However, evaluating $f(p_i)$ may be prohibitive for real-time applications since it requires many forward passes through the network, thus we proposed to use coarse (smaller) neural SDFs approximating $f$ for the early iterations.

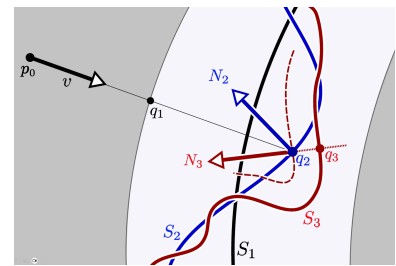

Specifically, let $f_1, f_2, f_3$ be neural SDFs with zero-level sets $S_1, S_2, S_3$ sorted by complexity (we give the definitions of $f_i$ in Sec 3.3), then to sphere trace $S_3$ we use $f_1$ and $f_2$ in the early iterations. For this, we need $S_3$ to be *nested* in a $\delta$-neighborhood of $S_2$, i.e. $S_3 \subset \big[|f_2| \leq \delta\big]$ (see Fig. 3).

Thus, we ray trace $f_2^{-1}(\delta)$ iterating $p_{i+1} = p_i + v\big(f_2(p_i) - \delta\big)$ and continue the iterations in the $\delta$-neighborhood using the target SDF $f_3$. Therefore, if the ray $p_0 + tv$ intersects $S_3$, the above procedure converges.

Moreover, to use $f_1$ we need an additional condition. To extend the above procedure to the sequence $f_i$, we should first sphere trace a coarser level set $f_1^{-1}(\delta_1)$, then, $f_2^{-1}(\delta_2)$, and finally, $S_3$. For such algorithm to converge, we need those neighborhoods to be *nested* as follows, otherwise, we may miss the hit point (see Fig. 4 (b)).

$$S_3 \subset \big[|f_2| < \delta_2\big] \subset \big[|f_1| < \delta_1\big] \qquad (1)$$

The choice of $\delta_1$ and $\delta_2$ values plays an important role on rendering. Having different values for them is also necessary to avoid issues as illustrated in Fig. 4. In Section 3.3 we present a definition for $\delta_i$ relating it with the network training.

In practice, we may choose how to use the SDFs $f_2$ and $f_3$ to adapt to a specific performance budget. We may choose to skip evaluating $f_2$, instead simply mapping the normals from $f_3$ directly onto $f_1$, thus decreasing the rendering cost. Iterating on $f_2$ while mapping normals from $f_3$ increases the cost, but its still cheaper than performing the full pipeline. Finally, iterating on all $f$'s presents the best silhouette results, although at a greater computational cost. Section 4.2 presents an evaluation of those cases.

Figure 2: *Multiscale ST*: to sphere trace $S_2$ we first sphere trace the boundary of a neighborhood of $S_1$ (gray), resulting in $q_1$. Then we continue to sphere trace $S_2$, reaching $q_2$. *Attribute mapping*: since $q_2$ belongs to a neighborhood of $S_3$, we evaluate the normal $N_3$ at $q_2$ of a parallel surface of $S_3$ (red dotted). These surfaces share the same normals. Color is acquired by making it constant along the line $q_3 + tN_3$.

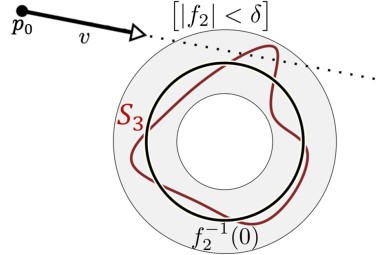

Figure 3: Ray intersecting $S_3$ nested in a $\delta$-neighborhood of a coarse SDF $f_2$. Notice that sphere tracing $f_2$ directly would lead to a false negative, thus we use $\big[|f_2| < \delta_2\big]$ instead.

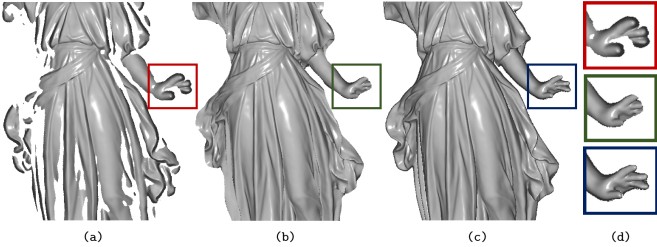

Figure 4: Implications of $\delta_i$ in ST, when the number of iterations are fixed. (a) Using too large $\delta_1 = \delta_2$ may result in holes (the ray do not reach the surface). More iterations would be needed using the finer (more complex) SDF to fill those holes, defeating the idea of minimizing iterations. (b) Conversely, reducing the deltas $\delta_1 = \delta_2$ may miss parts of the silhouette since the target surface may not be inside the previous neighborhood (notice the hand). (c) Using $\delta_1$ and $\delta_2$ suited for the nesting condition implies in no holes and a better silhouette capture.

### 3.3 TRAINING THE SDFs WITH NESTED NEIGHBORHOODS

This section describes approaches to define sequences of neural SDFs with nested neighborhoods. The objective is to train this sequence sorted by inference time and find small thresholds that ensure the nesting condition (1).

We define the coarse SDF $f_1$ as a small sinusoidal MLP which will be used at the first ST iterations. Then, we define the medium and fine SDFs $f_2, f_3$ simply using a residual scheme as follows

$$f_{i+1} = f_i + r_i, \quad \text{for } i = 1, 2. \tag{2}$$

In other words, to define a fine SDF $f_{i+1}$ we sum the coarser SDF $f_i$ with a residual sinusoidal MLP $r_i$ with a wider bandlimit. For this, we use the frequency parameter $\omega_0$ of SIREN. We refer to this sequence $f_i$ as *multiscale SDFs*.

We now define the nesting parameters $\delta_1$ and $\delta_2$ to enforce the SDFs $f_i$ to be nested during training, that is, (1) must hold. First, we train the base SDF $f_1$ in the whole domain $\Omega$, then we note that we can train $f_{i+1}$ for $i = 1, 2$ restricted to $\left[|f_i| < \delta_i\right]$ since the ST (see Alg. 1) do not evaluate $f_{i+1}$ outside this region.

Specifically, let $\{x_j, N_j\}_{j=1}^n$ be an oriented point cloud (the *ground-truth*) consisting of points $x_j$ and their normals $N_j$ sampled from a surface $S$. To train the INRs $f_i$ we follow the common approach of defining loss functions to enforce the Eikonal eq. $\mathcal{E}(f) := 1 - \|\nabla f\| = 0$ to be satisfied.

$$\mathcal{B}(f_1) = \underbrace{\frac{1}{n} \sum_j f_1(x_j)^2 + \left(1 - \langle \nabla f_1, N_j \rangle\right) + \int_\Omega \mathcal{E}(f_1)^2 dx}_{\mathcal{L}_{\text{data}}(f_1)}, \quad \mathcal{N}(f_i) = \mathcal{L}_{\text{data}}(f_i) + \int_{\left[|f_{i-1}| < \delta_{i-1}\right]} \mathcal{E}(f_i)^2 dx. \tag{3}$$

$\mathcal{B}$ is used to train the base SDF $f_1$ in the whole domain $\Omega$, while $\mathcal{N}$ only trains $f_2$ and $f_3$ on a $\delta_{i-1}$-neighborhood of the previously trained SDFs. This neighborhood training allows representing detailed SDFs $f_2$ and $f_3$ using small residual networks, see comparisons in Fig. 16. We define appropriate $\delta_i$ to enforce the sequence $f_i$ to satisfy the nesting condition (1). Precisely, we define $\delta_i$ such that $\left[|f_i| < \delta_i\right]$ contains the point cloud $\{x_j\}$ using the following formula.

$$\delta_i = (1 + \varepsilon) * \max_j |f_i(x_j)|. \tag{4}$$

Thus the training would force the zero-level set $f_{i+1}^{-1}(0)$ to approximate $\{x_j\}$ inside $\left[|f_i| < \delta_i\right]$ for $i = 1, 2$. In other words, the nesting condition would be satisfied by construction.

#### 3.3.1 SAMPLING THE GROUND-TRUTH SDF NEAR THE INPUT ORIENTED POINT CLOUD

In practice, to discretize the term $\int \mathcal{E}(f_i)^2 dx$ in $\left[|f_{i-1}| < \delta_{i-1}\right]$, we use an average on a dithering sampling around $\{x_j\}$ with a radius of $2\delta_{i-1}$. Then, we remove points outside the region using $f_{i-1}$. Fig. 5(a) depicts how this samplings works.

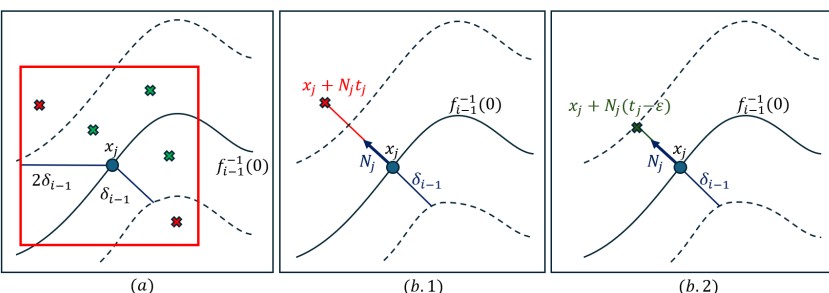

Figure 5: a) illustrates the dithering sampling for the Eikonal regularization of the SDF $f_i$. Points outside the $\delta_{i-1}$-neighborhood of $f_{i-1}$ (red) are removed. (b.1, b.2) illustrate the procedure for computing the displacement $t_j$ used to sample the ground-truth SDF near $x_j$. Starting with $t_j > \delta_{i-1}$, we iteratively decrease it by $\epsilon$ until the corresponding point falls within the neighborhood (green).

Additionally, we improve $\mathscr{L}_{\text{data}}$ for $i = 2, 3$ using the tubular neighborhood of $\{x_j\}$, see Fig. 5(b). Specifically, for each point $x_j$ we compute a number $t_j \leq \delta_{i-1}$ such that the distance of $x_j + tN_j$, with $t \in [0, t_j]$, to $\{x_j\}$ is exactly $t$. Observe that with $\{t_j\}$ in hands, we have $f_i(x_j + tN_j) = t$ and $\nabla f_i(x_j + tN_j) = N_j$ for each $t \in [0, t_j]$. Hence, during sampling we can also supervise the training of $f_i$ near the original point cloud.

We use an iterative approach to compute $\{t_i\}$. We start with $t_j = \delta_{i-1}$. Then, we compute the distance of $x_j + t_j N_j$ to $\{x_j\}$, if it is different from $t_j$ we replace $t_j$ by $t_j - \epsilon$; where $\epsilon > 0$ is a small number. The iteration ends when all $t_j$ do not need to be updated. Finally, note that the computation of $t_i$ can be performed as a preprocessing step and executed in parallel.

### 3.4 Multiscale Sphere Tracing

We propose the multiscale sphere tracing (Alg. 1), a variation of the classic algorithm to render multiscale SDFs $f_1$, $f_2$, $f_3$. Let $p$ be a point and $v$ be a direction, it approximates the first intersection (if it exists) between $S_3 = f_3^{-1}(0)$ and $\gamma(t) = p + tv$, with $t > 0$.

Specifically, we assume $p \notin \big[ |f_1| \leq \delta_1 \big]$. The multiscale ST is based on the fact that to sphere trace $S_3$ we can first sphere trace $f_1^{-1}(\delta_1)$ using $f_1$ (Fig. 3). Lines 3-6 describe the ST of $f_j^{-1}(\delta_j)$ for $j = 1, 2, 3$ (line 1). If $j = 3$ we sphere trace $S_3$ instead of its neighborhood (line 4).

---

**ALGORITHM 1:** Multiscale ST

**Input:** Sequence of nested neural SDFs $\{f_i\}$,
point $p$, direction $v$, threshold $\epsilon > 0$

**Output:** End point $p$

1 **for** $j = 1, 2, 3$ **do**
2    $t = +\infty$;
3    **while** $t > \epsilon$ **do**
4      $t = (j{==}3)\,?\,f(p) : f_j(p) - \delta_j$ ;
5      $p = p + tv$;
6    **end**
7 **end**

---

If $\gamma \cap S_3 \neq \emptyset$, the ST approximates the first hit point between $\gamma$ and $S_3$. This is due to the nesting condition, which ensures that if $\gamma \cap S_3 \neq \emptyset$ implies $\gamma \cap f_2^{-1}(\delta_2) \neq \emptyset$, and then $\gamma \cap f_1^{-1}(\delta_1) \neq \emptyset$.

For the inference of a neural SDF, in Line 4 of Alg. 1, we use the GEMM alg. (Dongarra et al., 1990) for each layer. To finish the rendering, we need to compute the normals and the textures.

### 3.5 Normal and texture mapping

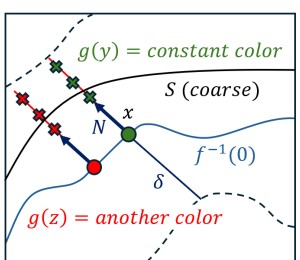

Figure 6: Volumetric texture mapping. The texture $g$ should be constant along the normals $N$ near the coarse surface $S$ (red/green). Having such volumetric representation in the $\delta$-neighborhood ensures that $g$ can be assigned to any point in the coarse surface $S$.

Let $S$ be a surface nested in a $\delta$-neighborhood of the zero-level set of a neural SDF $f$, that is, $S \subset \big[ |f| \leq \delta \big]$. Assume $f$ to be a finer neural SDF, then the *neural normal mapping* assigns to each $p \in S$ the attribute $g(p) := \nabla f(p)$. This is a restriction of $\nabla f$ to $S$ and maps the normal of $f^{-1}(0)$, along the minimum path connecting it to $p$. The attribute $g$ is constant along the path since $f$ is a SDF.

We explore two cases. First, let $S$ be a triangle mesh. We use the neural normal mapping to transfer the detailed normals of the level sets of $f$ to $S$. This approach is analogous to the classic normal mapping which depends on UV parameterizations. Since our method is volumetric, such parameterizations are not needed (see Fig. 9 - middle). For the second case, let $S$ be the zero-level set of another coarse neural SDF. We can use the neural normal mapping to avoid the overhead of additional ST iterations (see Fig. 9 - left). In this case, we do not need to extract a surface using marching cubes.

Similarly, we define a neural network $g : \mathbb{R}^3 \to \mathcal{C}$ to encode a *texture* on the $\delta$-neighborhood of $f$ with codomain $\mathcal{C}$ being the RGB space. We denote the attribute mapping associated to the triple $\{S, f, g\}$ a *neural texture mapping*. To train the parameters $\phi$ of $g$ we use the following loss functional: $\mathscr{T}(\phi) = \int_{f^{-1}(0)} (g - g)^2 dx + \int_{\big[|f| \leq \delta\big]} \langle \nabla g, \nabla f \rangle^2 dx$. where the first term forces $g$ to fit to the *ground-truth* texture $g$, and the second term asks for $g$ to be constant along the gradient paths, that is, it regularizes the network on the $\delta$-neighborhood of $f$. Fig. 6 depicts how the texture mapping works.

### 3.6 GEMM-BASED ANALYTICAL NORMAL CALCULATION FOR MLPs

We propose a GEMM-based analytical computation of normals, which are continuous and do not need auto-differentiation. This results in smooth normals, as shown in Fig. 7c. To compute the normals, we recall that a MLP with $n-1$ hidden layers has the following form:

$$f(x) = W_n \circ h_{n-1} \circ \cdots \circ h_0(x) + b_n, \qquad (5)$$

where $h_i(x_i) = \varphi(W_i x_i + b_i)$ is the $i$-layer. The *activation* $\varphi$ is applied on each coordinate of the linear map $W_i : \mathbb{R}^{N_i} \to \mathbb{R}^{N_{i+1}}$ translated by $b_i \in \mathbb{R}^{N_{i+1}}$. The gradient of $f$ is given using the *chain rule*:

$$\nabla f(x) = W_n \cdot \mathbf{J}h_{n-1}(x_{n-1}) \cdots \cdot \mathbf{J}h_0(x), \quad \text{with} \quad \mathbf{J}h_i(x_i) = W_i \odot \varphi' \big[a_i | \cdots | a_i\big] \qquad (6)$$

$\mathbf{J}$ is the *Jacobian*, $x_i := h_{i-1} \circ \cdots \circ h_0(x)$, $\odot$ is the *Hadamard* product, and $a_i = W_i(x_i) + b_i$. Eq. 6 is used in (Gropp et al., 2020; Novello et al., 2022) to compute the level set normals analytically.

We now use Eq. 6 to derive a GEMM-based algorithm for computing the normals ($\nabla f$) in real-time. The gradient $\nabla f$ is given by a sequence of matrix multiplications which is not appropriate for a GEMM setting because $\mathbf{J}h_0(x) \in \mathbb{R}^{3 \times N_1}$. The GEMM algorithm organizes the input points into a matrix, where its lines correspond to the points and its columns organize them and enable parallelism. We can solve this problem using three GEMMs, one for each normal coordinate. Therefore, each GEMM starts with a column of $\mathbf{J}h_0(x)$, eliminating one of the dimensions. The resulting multiplications can be asynchronous since they are completely independent.

The $j$-coord of $\nabla f$ is given by $G_n = W_n \cdot G_{n-1}$, where $G_{n-1}$ is given by iterating $G_i = \mathbf{J}h_i(x_i) \cdot G_{i-1}$, with the initial condition $G_0 = W_0[j] \odot \varphi'(a_0)$. The vector $W_0[j]$ denotes the $j$-column of $W_0$. We use a kernel and a GEMM to compute $G_0$ and $G_n$. For $G_i$ with $0 < i < n$, observe that

$$G_i = (W_i \odot \varphi' [a_i | \cdots | a_i]) \cdot G_{i-1} = (W_i \cdot G_{i-1}) \odot \varphi'(a_i).$$

The first equality comes from Eq. 6 and the second from a commutative property of the Hadamard product. The second expression needs fewer computations and is solved using a GEMM followed by a kernel. Please refer to Appx A.2 for a detailed algorithm.

## 4 EXPERIMENTS

We compare our framework against SOTA methods, present ablation studies and additional applications. For the sphere tracing related experiments, we fix the number of iterations for better control of parallelism. All experiments are conducted on an NVidia RTX 3090.

We use a notation to refer to the MLPs: $(N, d)$ means a MLP with $d$ hidden layers of the form $\mathbb{R}^N \to \mathbb{R}^N$. Additionally, $(64, 1) \triangleright (256, 1)$ means a multiscale SDFs with a MLP with two hidden layers $\mathbb{R}^{64} \to \mathbb{R}^{64}$, and a MLP with two hidden layers $\mathbb{R}^{256} \to \mathbb{R}^{256}$.

| Neural Armadillo | Training (s) |
|---|---|
| (64, 1) (base) | 23.6 |
| (128, 1) (residual) | 40.2 |
| (256, 1) (residual) | 85.1 |
| Total | **148.9** |
| IDF | **100.1** |
| NGLOD | 1628.0 |

Table 1: Although our method is real time for rendering, its training time is comparable to IDF which depends on marching cubes to render, losing the smoothness of INRs. Our training is one order of magnitude faster than NGLOD.

### 4.1 COMPARISONS

**Surface:** First, we compare our neighborhood-nesting approach with SOTA methods for surface representation. The first one is implicit displacement fields (IDF)(Wang et al., 2022), which disentangles shape and detail. The second one is NGLOD(Takikawa et al., 2021), which is the only real-time rendering method that uses neural SDFs. Tab. 1 compares the training times. Even though our rendering is real-time, we have comparable training times against IDF, which rely on mesh extraction for rendering. Our training is one order of magnitude faster than NGLOD.

Figs. 7 and 15 show rendering comparisons. 7a uses the real-time configuration for NGLOD, recommended by the authors in their code repository. As discussed in Sec. 2, its formulation results in non-continuous normals, causing discretization artifacts. To increase geometric details using NGLOD, we have to consider a non-real-time LOD 5 configuration (7b), which has less discretization artifacts. 7c shows our real-time rendering framework. Since our approach works on the smooth setting, we support smooth normals. 7d shows the surface generated by IDF, after a marching cubes extraction of resolution $512^3$. Note that IDF and NGLOD do not support textures.

**Normals:** We compare our GEMM normal calculation with `torch.autograd`. As shown in Tab. 5, ours performs $2\times$ faster. We tested 6 different INRs trained for Armadillo, Happy Buddha, and Lucy, varying between 2-3 hidden layers.

Figure 7: Render comparison.

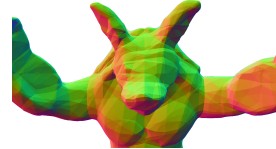

(a) NGLOD LOD 0 (real-time). Note the discretization artifacts (mosaic appearance).

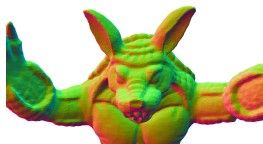

(b) NGLOD LOD 5 (not real-time). Less artifacts.

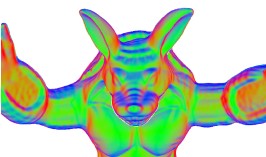

(c) Ours, with configuration $(64, 1) \triangleright (128, 1) \triangleright (256, 1)$ (real-time). Note the smooth normals.

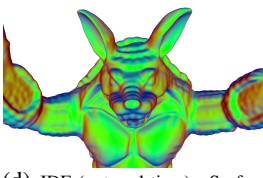

(d) IDF (not real-time). Surface extracted using marching cubes.

**Textures:** Since our approach is the first to address textures for neural SDFs in real-time, we present a comparison against classical uv-textures on meshes. We present the MSE between the images generated by our method and the texture meshes. We consider the models: Spot, Bob, Bunny, Egg, and Earth. The corresponding MSEs are: 0.0329, 0.0434, 0.0720, 0.0291, and 0.0033. Please refer to Fig. 17 (Appx. A.3) for the images used to compute the MSEs. Fig. 8 shows the neural texture mapping applied to coarse surfaces.

Since our method defines the textures in a neighborhood of the surface, no parameterization or uv-map is needed. Inference is simple and consists of a single MLP evaluation for a batch of points. The results show that our approach achieves good appearance while uncoupling it from geometry in a compositional manner.

### 4.2 ABLATION STUDIES

**Residuals:** First, we evaluate the impact of the residual approach. Fig. 10 shows that residuals eliminate spurious components when applied to neighborhood training. We use this property to accelerate marching cubes in the case mesh extraction is needed.

To evaluate the efficiency of coarse neural SDFs to represent the ground-truth SDFs, Tab. 2 shows the Hausdorff distances between their zero-level sets and the original point clouds. All distances are within the third decimal digit, which means they are very close to the ground-truth. This fact corroborates our assumption that coarse surfaces in nested neighborhoods can be used to accelerate rendering.

**Neural normal mapping and multiscale ST:** Regarding image quality and perception, Fig. 9 shows the case where the coarse surface is the zero-level of a neural SDF (left) and when it is a triangle mesh (middle), showing that our representation can also be beneficial for rendering meshes. An overall evaluation of the algorithm with other models is given in Fig. 16 (Appx. A.3). In all cases, normal mapping increases fidelity.

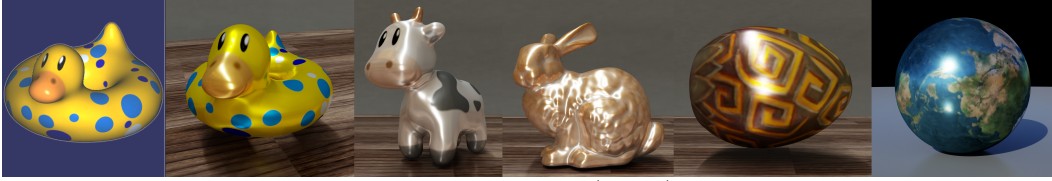

Figure 8: Neural texture mapping. All networks are $(256, 3)$, except for the the earth, which is $(512, 3)$. The first case on the left is a sphere traced surface. The other cases are marching cubes of $(64, 1)$ SDFs, except for the bunny, which is $(128, 2)$. No parameterization or uv-map is needed.

| Model | Nets | Dist. |
|-------|------|-------|
| Arm. | (64,1) | 0.0035 |
|  | (256,3) | 0.0021 |
| Bunny | (64,1) | 0.0024 |
|  | (256,1) | 0.0019 |
|  | (256,3) | 0.0021 |
| Buddha | (64,1) | 0.0051 |
|  | (256,1) | 0.0019 |
|  | (256,3) | 0.0016 |
| Lucy | (64,1) | 0.0071 |
|  | (256,1) | 0.0024 |
|  | (256,3) | 0.0017 |

Table 2: Hausdorff distance between the trained models and the ground-truth.

The result may be improved using the multiscale ST, as shown in Fig. 9 (right). Adding ST iterations using a neural SDF with a better approximation of the surface improves the silhouette (right).

**Real-time renderer:** We evaluate a GPU version implemented in a CUDA renderer, using neural normal mapping, multiscale ST, and the GEMM-based analytical normal calculation (implemented using CUTLASS). Tab. 3 shows the results. Notice that the framework achieves real-time performance and that using neural normal mapping and multiscale ST improves performance considerably. An ablation study varying the number of sphere tracing iterations per level of detail is presented in Tab. 6 (Appx.).

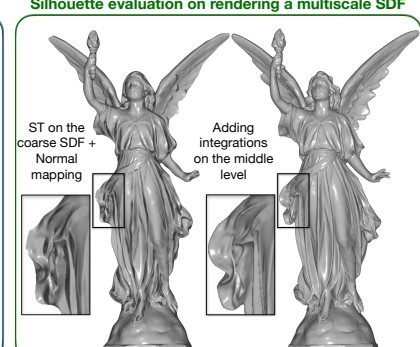

Figure 9: Left: neural normal mapping onto a neural SDF. First, the coarse $(64, 1)$ SDF. Then, the neural normal mapping of the $(256, 3)$ SDF onto the $(64, 1)$. Middle: neural normal mapping onto half of a triangle mesh. The normals of the $(256, 3)$ SDF are used. The mesh is the marching cubes of the $(64, 1)$ SDF. The *mean square error* (MSE) is 0.00262 for the coarse case and 0.00087 for the normal mapping, an improvement of $3\times$. The baseline is the marching cubes of the $(256, 3)$ SDF. Right: Silhouette evaluation. First a $(64, 1) \triangleright (256, 3)$, then a $(64, 1) \triangleright (256, 2) \triangleright (256, 3)$ configuration. Notice how the silhouette improves with the additional $(256, 2)$ level.

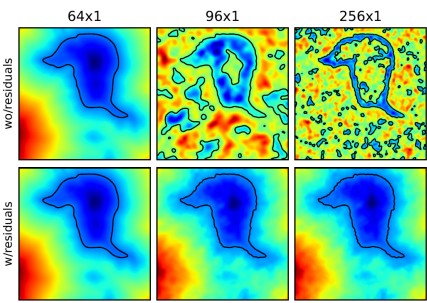

Figure 10: Evaluation of the residual approach. Note that training the SDFs in the neighborhoods (first row: center, right) results in spurious components outside the neighborhood as would be expected. Using the residual approach eliminates those components (second row: center, right).

| Model | FPS | Speedup | Size |
|---|---|---|---|
| $(256, 3)$ (SIREN baseline) | 19.8 | 1.0X | 777 |
| NGLOD (real-time) | 20.0 | 1.0X | 96 |
| $(64, 1)$ (coarse) | **124.1** | **6.3X** | 18 |
| $(64, 1) \triangleright (128, 1)$ (res, NM) | **80.0** | **4.0X** | 86 |
| $(64, 1) \triangleright (128, 1) \triangleright (256, 1)$ (res, NM) | **41.2** | **2.1X** | **349** |
| $(64, 1) \triangleright (128, 1) \triangleright (256, 1)$ (res) | **32.2** | **1.6X** | **349** |
| $(64, 1) \triangleright (256, 2)$ (NM) | **70.4** | **3.6X** | 538 |
| $(64, 1) \triangleright (256, 2) \triangleright (256, 3)$ (res, NM) | **40.4** | **2.0X** | 1315 |
| $(64, 1) \triangleright (256, 2) \triangleright (256, 3)$ (res) | **31.7** | **1.6X** | 1315 |

Table 3: Real-time evaluations using multiscale ST, GEMM normals, and normal mapping, in a CUDA renderer. The number of iterations is 20 for the first neural SDF and 5 for the subsequent ones. (NM) indicates normal mapping of the last SDF and (res) indicates the residual approach. Images are $512^2$. Size is in KB. Note that the residual approach allows smaller networks and that all cases result in speedups. Although NGLOD runs at an average of 20 FPS, its underlying INR cannot represent fine geometric details in such setting, see Fig. 7(a).

## 4.3 ADDITIONAL APPLICATIONS

The flexibility of our multiscale SDF representation enables additional applications, including integration into differentiable pipelines and fast mesh extraction using the marching cubes algorithm.

**Neural implicit surface evolution**. Note that neural SDFs provide a smooth representation of a static scene. By adding an additional input coordinate, we can encode time into the representation. We leverage this approach to train dynamic evolutions of static

| Model | Baseline | No culling | Culling | Speedup |
|---|---|---|---|---|
| Arm. | 10.385 | 8.234 | **2.177** | **4.7×** |
| Buddha | 10.384 | 8.195 | **1.913** | **5.4×** |
| Lucy | 10.404 | 8.410 | **1.481** | **7.0×** |

Table 4: Average marching cubes runtime comparison in seconds. Note that we averaged the runtime of 100 runs, while discarding the first, used as a warmup. Surface reconstructions are shown in Fig. 18, in the Appx.

neural SDFs, following the training schemes introduced in (Novello et al., 2023). Fig. 11 presents an example of interpolation between the Spot and Bob models using this method. Importantly, the implicit model handles topology changes, demonstrating that our representation can be integrated into differentiable pipelines. The visualization is in real-time (120 FPS) using an extension of our multiscale ST to dynamic SDFs.

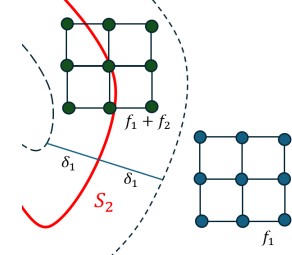

(a)   (b)   (c)   (d)   (e)

Figure 11: A dynamic multiscale SDF is trained using the pipeline from (Novello et al., 2023). Note the change in topology (c-d), which is challenging to handle using meshes. Also, octree/mesh-based approaches require generating a surface for each time, an overhead that our model avoids.

**Residuals remove spurious components.** This property is a direct consequence of our residual approach. Fig. 10 shows that residuals eliminate spurious components when applied to neighborhood training. We use this property to accelerate marching cubes in case of mesh extraction.

**Improving Marching Cubes performance.** We can use our multiscale SDF to speed up mesh extraction. Experiments show that our representation improves the performance of grid evaluation, by avoiding inference at finer levels for vertices far from the zero-level set. The key idea is to use the coarse version of the neural SDF for grid vertices culling. Only those in the nesting neighborhood of the coarse surface use finer SDF, enabling an adaptive sampling of SDF values. Fig. 12 shows the approach, Fig. 18 (Appx.) shows the surface reconstructions, and Tab. 4 demonstrates a maximum performance improvement of $7\times$. For all cases, the baseline SDF is approximated by a single MLP with configuration $(256, 3)$, while the multiscale SDFs have configuration of $(64, 1) \triangleright (128, 1) \triangleright (256, 1)$. Note that surfaces occupying smaller domain region have a greater speed up since the number of vertices on their nesting neighborhoods decrease.

Figure 12: Adaptive marching cubes. For grid vertices outside the $\delta_1$ neighborhood (blue), only the coarse SDF $f_1$ is evaluated. For points in the neighborhood (green) the residual $f_2$ is added.

**Reconstruction from (noisy) point-clouds.** Our method may also be integrated into surface reconstruction pipelines, allowing the use of potentially noisy point-clouds as input. See Fig. 13 in the Appx. for details.

## 5   CONCLUSION

We propose an end-to-end INR framework to render surfaces in real-time using smooth neural SDFs endowed with smooth attributes such as normals and textures. It leverages on spatial neighborhoods and residual training, achieving real-time performance without the need of spatial data structures. The multiscale ST accelerates the surface evaluation, the neural attribute mapping transfers surface attributes from a neural SDF to another surface, and the GEMM-based analytical normal computation provides smooth normals without the need of auto-differentiation. Moreover, we demonstrate that our multiscale neural SDF can be easily adapted to differentiable, time-dependent pipelines for surface evolution. Additionally, we leverage the nesting neighborhood to accelerate mesh extraction using marching cubes.

| Model | Autograd | Ours | Resolution |
|---|---|---|---|
| Armadillo 256x2 | 0.007 | **0.003** | 512x512 |
| Armadillo 256x2 | 0.024 | **0.010** | 1024x1024 |
| Armadillo 256x3 | 0.010 | **0.005** | 512x512 |
| Armadillo 256x3 | 0.025 | **0.012** | 1024x1024 |
| Buddha 256x2 | 0.008 | **0.005** | 512x512 |
| Buddha 256x2 | 0.021 | **0.014** | 1024x1024 |
| Buddha 256x3 | 0.011 | **0.005** | 512x512 |
| Buddha 256x3 | 0.024 | **0.012** | 1024x1024 |
| Lucy 256x2 | 0.007 | **0.004** | 512x512 |
| Lucy 256x2 | 0.021 | **0.012** | 1024x1024 |
| Lucy 256x3 | 0.011 | **0.007** | 512x512 |
| Lucy 256x3 | 0.025 | **0.015** | 1024x1024 |

Table 5: Runtime comparison, in seconds, between Pytorch autograd and our algorithm to calculate the normals. Ours performs $2\times$ faster.

**Limitations and future work.** As common for SDF-based representations, our approach is not suitable for representing sharp edges. This is a natural consequence of the function smoothness and may be solved by incorporating local features into the function, a path we would like to explore in future work. The multiscale ST could probably be applied into neural SDF-based 3D reconstruction or inverse rendering tasks to reduce the training time. Nested neighborhoods could be adapted for unsigned distance functions too. Improvements can be done for further performance optimization. For example, using fully fused GEMMs may decrease the overhead of GEMM setup (Müller, 2021).

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

# A APPENDIX

## A.1 ON THE IMPORTANCE OF SMOOTH REPRESENTATIONS AND REAL-TIME RENDERING

This section clarifies why the smoothness and real-time rendering of our multiscale INR are important, and presents a contextualization with triangle meshes showing applications where the mentioned properties are essential.

Our objective is not to replace meshes, in fact, we show that using our multiscale INR allows fast mesh extraction using marching cubes (Section 4.3). Rather, we provided an implicit representation for SDFs suitable for specific tasks depending on two main properties: 1) **high order differentiability** (smoothness); 2) **fast level set rendering** (especially when surface extraction may be prohibitive).

For an example, Section 4.3 gives an application of our method on surface evolution using differential equations that explores those two properties. Specifically:

**Smoothness.** We use the method Neural implicit surface evolution (Novello et al., 2023) (NISE) which trains dynamic SDFs using the level-set method. Thus, in addition to the Eikonal regularization necessary for the SDF, we need higher derivatives to compute differential properties (e.g. mean curvature). For this, we use the smoothness our INR, making its integration with NISE easier. Conversely, using meshes for surface evolution is challenging because the representation should be adapted to handle the lack of differentiability. Finally, meshes cannot easily handle topology changes (eg. the Spot-Bob interpolation shown in Figure 11). Creating holes in the mesh during animation is a hard task due to its fixed topology. This problem is easily avoided using our implicit representation.

**Fast rendering.** To visualize the resulting animation, the zero-level sets must be evaluated fast during evolution for real-time rendering. This is achieved by integrating our multiscale INR with NISE. On the other hand, using meshes would be prohibitive because the mesh should be extracted for each time instant of the animation, which cannot be done in real-time. Preprocessing the animation is also unfeasible since each mesh extraction may take dozens of megabytes (see the comments about mesh extraction below), creating an unacceptable memory footprint. Using our sphere tracing we only need to store the underlying MLP.

Finally, an additional objective of providing real-time rendering for neural SDFs is making its integration in neural pipelines more appealing. For example, fast rendering of such INRs is useful in inverse rendering tasks since it helps accelerate training. Previous works that propose such pipelines include DIST (Liu et al., 2020) and SDFDiff (Jiang et al., 2020). Additionally, SDF is popular surface representation in 3D reconstruction from images using differentiable volume rendering (eg. NeuS (Wang et al., 2021) and volSDF Yariv et al. (2021)).

**Mesh extraction and SDF training as pre-processing for rendering.** Extracting a mesh from a trained neural SDF results in a substantial memory footprint, especially when the zero-level set is highly detailed. This is primarily due to the cubic complexity of grid generation for marching cubes. For example, we trained a multiscale SDF in our experiments using the following architecture: $(128, 1)$ for coarse level, $(256, 1)$ for medium level, and $(256, 2)$ for fine level. Generating the grid of resolution $512^3$ and running the marching cubes for this case demands approximately 20 GB of GPU memory while rendering with our sphere tracing using an image resolution of $512^2$ requires significantly less—approximately 5 GB, including the GEMM buffers used to parallelize the pixel computation. Additionally, storing high-resolution meshes is costly in terms of memory. For this experiment, the output mesh has 43 MB of storage, while the underlying multiscale MLP representation needs only 857 KB, showing that our representation is significantly more compact.

## A.2 GEMM-BASED NORMAL COMPUTATION ALGORITHM

Algorithm 2 presents the gradient computation for a batch of points as described in Section 3.6. The input is a matrix $P \in \mathbb{R}^{3 \times k}$ with columns storing the $k$ points generated by the GEMM version of Algorithm 1. The algorithm outputs a matrix $\nabla f_\theta(P) \in \mathbb{R}^{3 \times k}$, where its $j$-column is the gradient of $f_\theta$ evaluated at $P[j]$. Lines $2-5$ are responsible for computing $G_0$, Lines $6 - 11$ compute $G_{n-1}$, and Line 13 provides the result gradient $G_n$. Table 5 shows a comparison between this algorithm and automatic differentiation using pytorch.

## A.3 ADDITIONAL EXPERIMENTS

**Point cloud from images:** Fig 13 shows our model trained with a point cloud reconstructed from an image. We use Depth Anything (Yang et al., 2024) to generate the depth of the pixels and use that depth to create the point cloud based on the view.

**More complex shapes:** Fig. 15 shows a comparison of our representation against NGLOD (Takikawa et al., 2021) and IDF Wang et al. (2022) in the complex Asian Dragon shape. We achieve a fidelity near IDF in a real-time context.

---

**ALGORITHM 2:** Normal computation

**Input:** neural SDF $f_\theta$, positions $P$
**Output:** Gradients $\nabla f_\theta(P)$

1  **for** $j = 0$ *to* $2$ *(async)* **do**
2      using a GEMM: `// Input Layer`
3          $A_0 = W_0 \cdot P + b_0$
4      using a kernel:
5          $G_0 = W_0[j] \odot \varphi'(A_0)$;
        $P_0 = \varphi(A_0)$
    `// Hidden layers`
6      **for** *layer* $i = 1$ *to* $n - 1$ **do**
7          using GEMMs:
8              $A_i = W_i \cdot P_{i-1} + b_i$;
            $G_i = W_i \cdot G_{i-1}$
9          using a kernel:
10             $G_i = G_i \odot \varphi'(A_i)$;
            $P_i = \varphi(A_i)$
11     **end**
12     using a GEMM: `// Output layer`
13         $G_n = W_n \cdot G_{n-1}$
14 **end**

---

**Integration with NeuS (Wang et al., 2021):** NeuS employs an implicit representation based on SDFs to accurately reconstruct 3D surfaces. The SDF gradient plays a crucial role in defining surface normals, smoothly calculating volumetric density, and ensuring geometric consistency through Eikonal regularization. It is also utilized to guarantee smooth transitions between geometry and density during volumetric rendering, enabling the modeling of detailed and complex surfaces. By combining these techniques with differentiable rendering, NeuS offers an efficient and robust approach for tasks such as surface reconstruction and 3D texture generation. Summing the gradients of two SDF networks, guided by our loss function, enhances NeuS by enabling a multi-scale representation and improving stability. One network can model large-scale structures, while the other captures fine details, with their combined gradients seamlessly integrating these scales for a more comprehensive representation. This multi-scale approach ensures better handling of complex geometries, particularly in challenging scenarios like thin structures or noisy inputs. Furthermore, the combined gradient mitigates irregularities and noise, leading to a smoother and more stable representation that better satisfies regularization constraints, such as the Eikonal condition, ultimately improving reconstruction quality, training robustness, and reducing processing time.

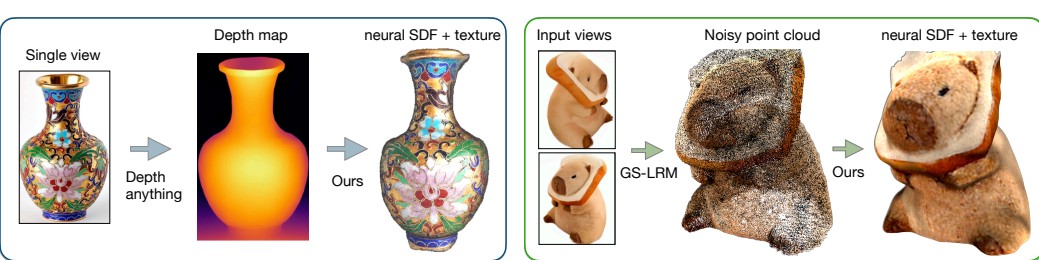

Figure 13: Training a textured SDF from images/noisy point cloud. On the left, our model (neural SDF + texture) is trained using the unprojection of a depth map, which is computed from a single view using Depth Anything. The resulting vase is rendered at 32.1 FPS. On the right, we show a reconstruction derived from a noisy point cloud, extracted from multiple views using GS-LRM (Zhang et al., 2024). By combining our method with this feed-forward 3D model (GS-LRM), we achieve fast reconstruction of the SDF with texture.

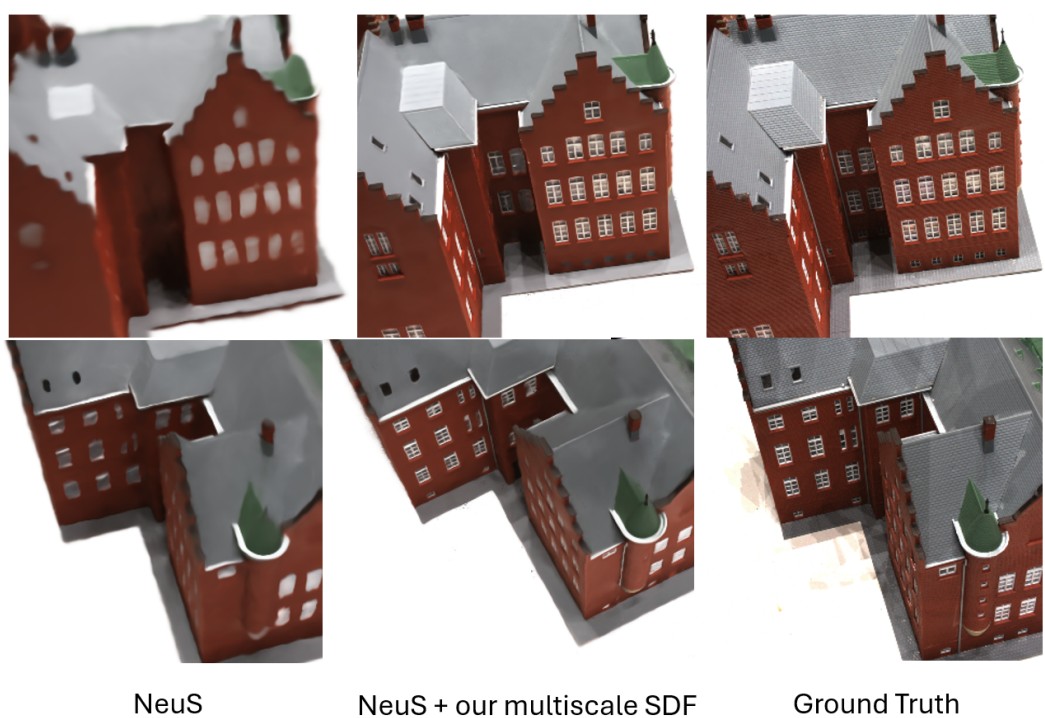

NeuS        NeuS + our multiscale SDF        Ground Truth

Figure 14: Integration of our multiscale representation SDF with NeuS (Wang et al., 2021). We compared the results of the first 20k iterations out of 300k total training iterations against a baseline NeuS model with 8 layers and 256 neurons in Fig. 14. Our approach demonstrates the ability to capture finer details even in the initial iterations.

Our loss function, when applied to the NeuS model, was able to generate detailed surfaces in just a few epochs of training. We trained a coarse model with 8 MLP layers and 64 neurons and a residual network, also with 8 MLP layers but 128 neurons. Both models follow the architecture of IDR (Yariv et al., 2020). We compared the results of the first 20k iterations out of 300k total training iterations against a baseline NeuS model with 8 layers and 256 neurons in Fig. 14. Notably, our approach, which combines the gradients of the two MLP networks, demonstrates the ability to capture finer details even in the initial iterations.

**Broader perceptual evaluation:** On the paper we exemplify results using one model for each experiment. Fig. 16 shows a broader perceptual evaluation of the multiscale sphere tracing and the neural normal mapping using several models. Fig. 17 also shows the images we use to calculate the MSE to compare the neural texture mapping with the rendering baseline.

**Accelerated Marching Cubes qualitative evaluation:** Fig. 18 shows high-fidelity reconstructions computed using our acceleration for the marching cubes algorithm.

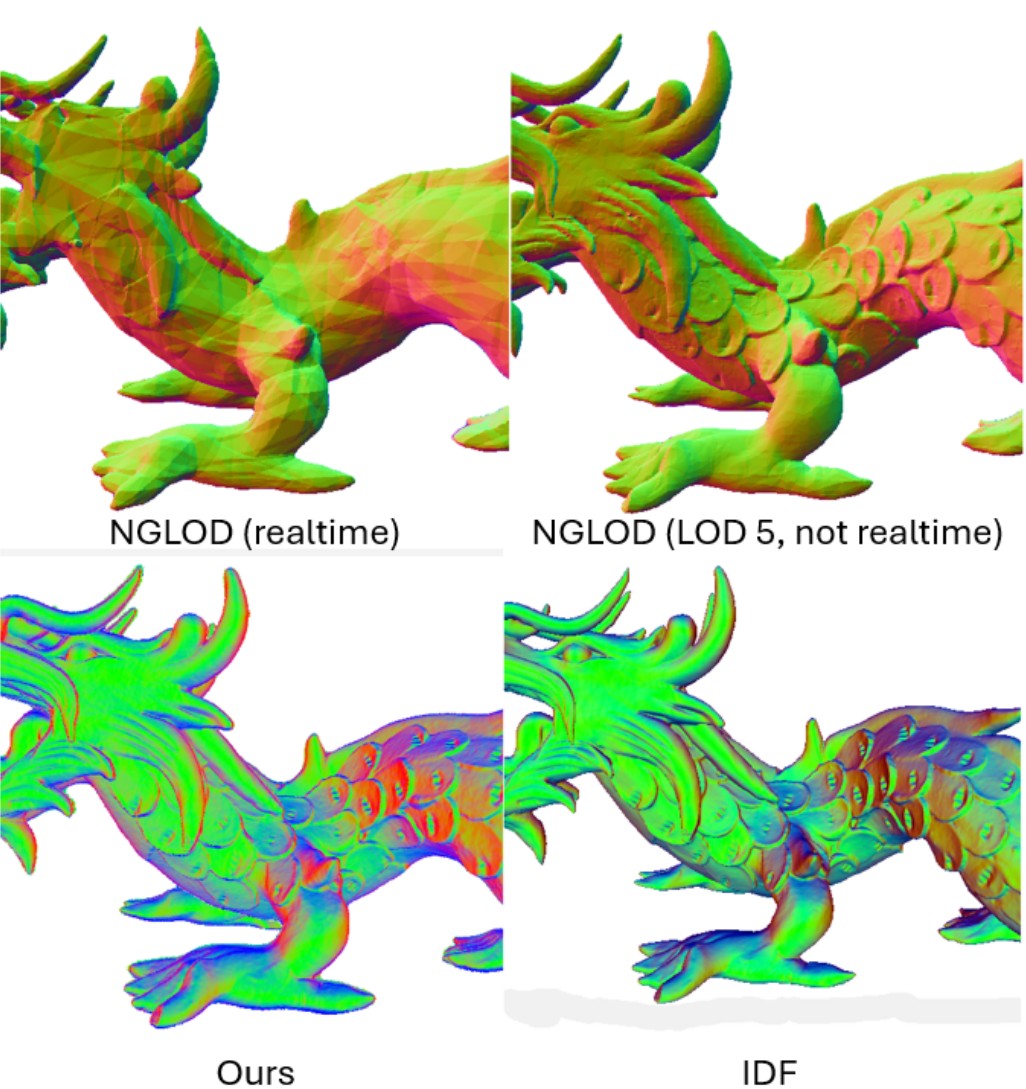

Figure 15: Comparison of our representation against NGLOD (Takikawa et al., 2021) (the reference real-time approach) and IDF Wang et al. (2022) (the reference non-real-time fidelity approach). We achieve high-fidelity real-time performance even for more complex shapes. Notice that our representation is able to learn the fine details of the dragon scales.

**Coarse**  **Normal mapping**  **Multiscale ST**  **Baseline**

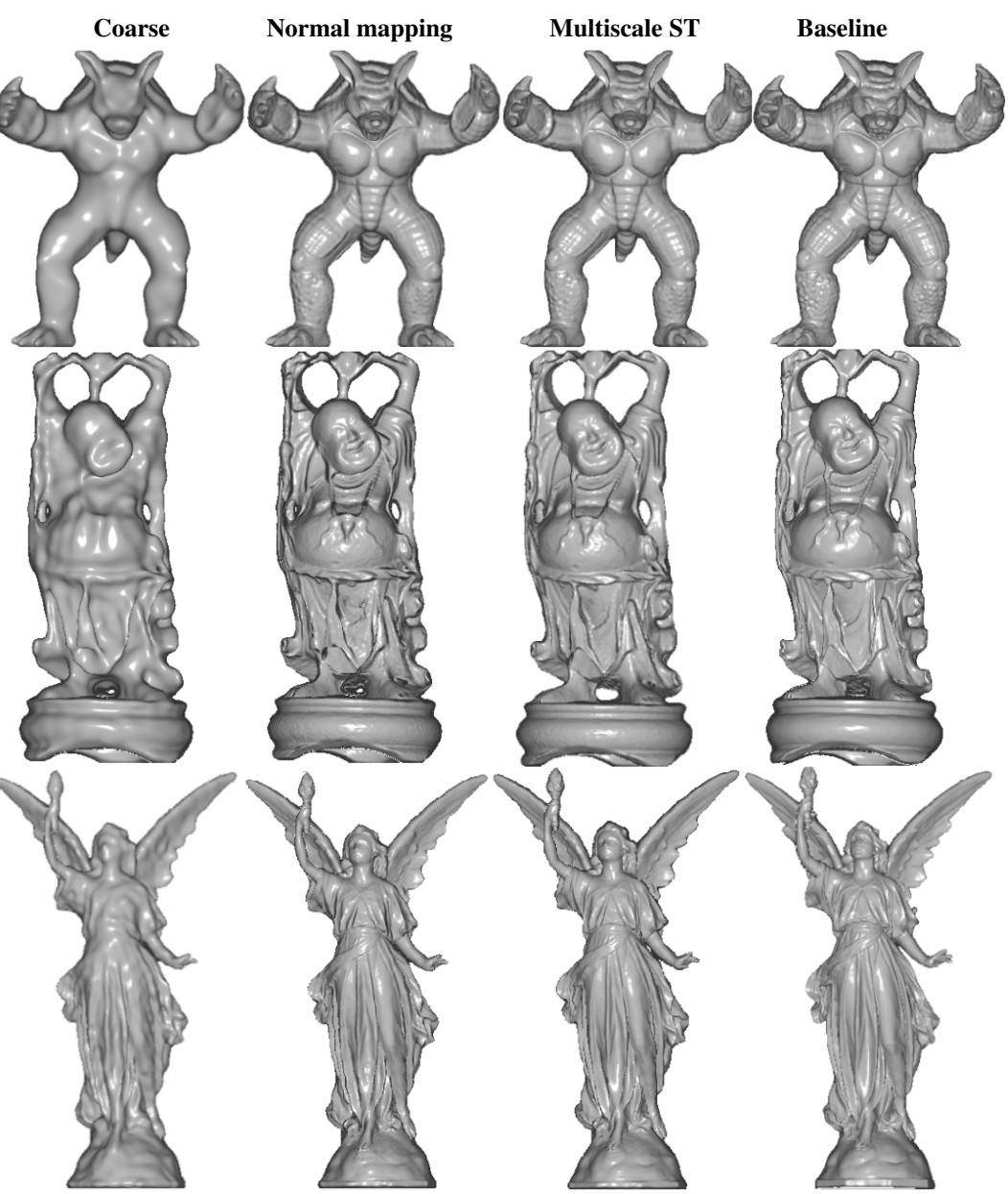

Figure 16: Comparison between our method and the SIREN baseline. The columns represent different configurations. From left to right: $(64, 1)$, $(64, 1) \triangleright (256, 1)$, and the baseline $(256, 3)$. The second column uses neural normal mapping and the third uses multiscale sphere tracing. Notice that fidelity is improved in the second column and the third column refines the results.

**Baseline**                    **Ours**

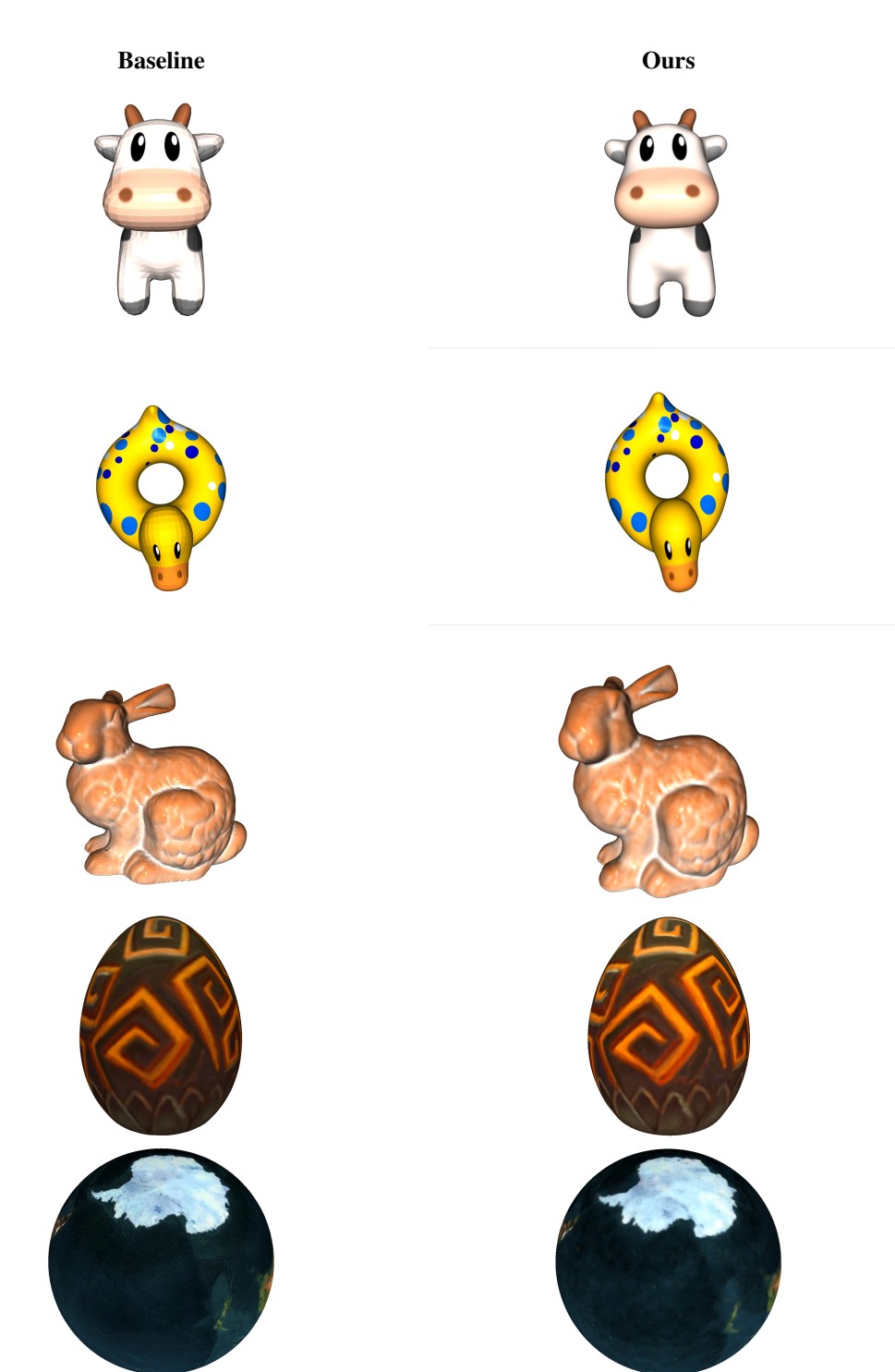

Figure 17: Images we use to calculate the MSE between the ground-truth textured meshes and our approach.

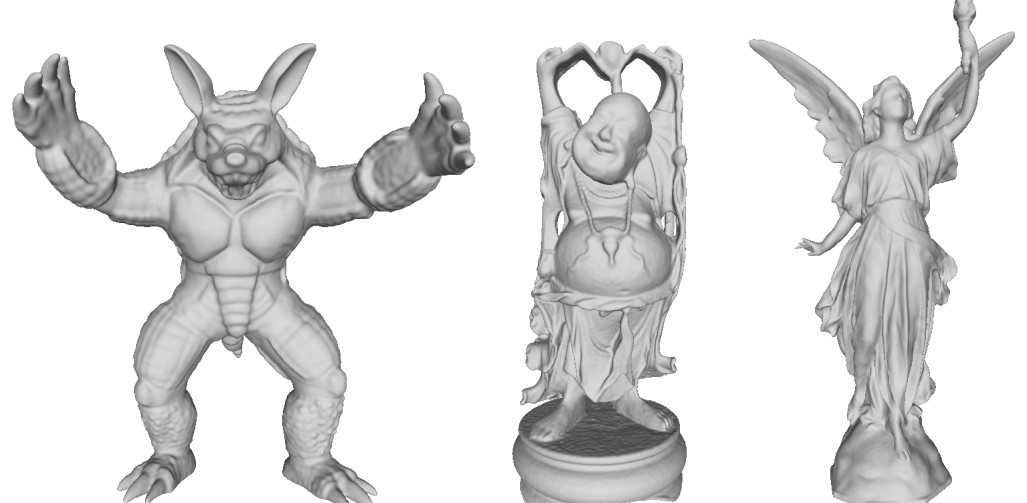

Figure 18: From left to right: Marching cubes reconstruction of Armadillo, Buddha and Lucy using our proposed grid culling method.

## A.4 ABLATION STUDIES

We performed three additional ablation studies for our approach: (i) detail level influence on time performance, (ii) loss term assessment, (iii) $\delta$ influence over the reconstructions. Table 6 shows the impact of sphere tracing iterations in time performance for all detail levels. It shows that our approach may used in a variety of performance budgets. Tables 7,8, and 9 show the ablation results of our loss function using different weights for each component, while maintaining the remaining hyper-parameters fixed. We performed these studies both for a single intermediate level (medium) and an additional refinement level beyond it (fine). Note that all studies used the Lucy mesh as a baseline. Table 10 shows the results for varying the delta values while maintaining the remaining hyper-parameters fixed.

| $f_1$ | $f_2$ | $f_3$ | FPS | $f_1$ | $f_2$ | $f_3$ | FPS | $f_1$ | $f_2$ | $f_3$ | FPS |
|---|---|---|---|---|---|---|---|---|---|---|---|
| 20 | 0 | 0 | 127.7 | 20 | 20 | 0 | 48.1 | 20 | 20 | 20 | 20.8 |
| 30 | 0 | 0 | 102.2 | 30 | 30 | 0 | 33.9 | 30 | 30 | 30 | 14.2 |
| 40 | 0 | 0 | 84.3 | 40 | 40 | 0 | 27.7 | 40 | 40 | 40 | 10.9 |
| 50 | 0 | 0 | 71.1 | 50 | 50 | 0 | 22.0 | 50 | 50 | 50 | 8.8 |

Table 6: Ablation study of the performance impact of sphere tracing iterations. $f_1$, $f_2$, and $f_3$ columns represent the number of sphere tracing iterations for the coarse, medium, and fine SDF respectively. FPS (frames per second) columns are the average of runs with several different $\delta_1$ and $\delta_2$ values.

| Gradient constraint | Approx. Error | | Gradient Constraint | Approx. Error |
|---|---|---|---|---|
| 0.0 | 0.0013 | | 0.0 | 0.0086 |
| 10.0 | 0.0013 | | 10.0 | 0.0084 |
| 30.0 | 0.0013 | | 30.0 | 0.0082 |
| 100.0 | 0.0012 | | 100.0 | 0.0078 |
| 300.0 | 0.0013 | | 300.0 | 0.0074 |
| 1000.0 | 0.0014 | | 1000.0 | 0.0069 |
| 3000.0 | 0.0017 | | 3000.0 | 0.0073 |
| 10000.0 | 0.0022 | | 10000.0 | 0.0087 |
| 30000.0 | 0.0030 | | 30000.0 | 0.0116 |

| (a) Gradient constraint fine level | (b) Gradient constraint medium level |
|---|---|

Table 7: Gradient constraint ablation studies.

| Normal Constraint | Approx. Error |
|---|---|
| 0.0 | 0.0017 |
| 10.0 | 0.0013 |
| 30.0 | 0.0013 |
| 100.0 | 0.0013 |
| 300.0 | 0.0013 |
| 1000.0 | 0.0013 |
| 3000.0 | 0.0013 |
| 10000.0 | 0.0013 |
| 30000.0 | 0.0013 |

(a) Normal constraint fine level.

| Normal Constraint | Approx. Error |
|---|---|
| 0.0 | 0.0073 |
| 10.0 | 0.0074 |
| 30.0 | 0.0077 |
| 100.0 | 0.0081 |
| 300.0 | 0.0083 |
| 1000.0 | 0.0085 |
| 3000.0 | 0.0086 |
| 10000.0 | 0.0087 |
| 30000.0 | 0.0087 |

(b) Normal constraint medium level.

Table 8: Normal constraint ablation studies.

| SDF Constraint | Approx. Error |
|---|---|
| 0.0 | 0.0076 |
| 10.0 | 0.0013 |
| 30.0 | 0.0013 |
| 100.0 | 0.0013 |
| 300.0 | 0.0013 |
| 1000.0 | 0.0012 |
| 3000.0 | 0.0013 |
| 10000.0 | 0.0013 |
| 30000.0 | 0.0013 |

(a) SDF constraint fine level.

| SDF Constraint | Approx. Error |
|---|---|
| 0.0 | 0.0490 |
| 10.0 | 0.0080 |
| 30.0 | 0.0079 |
| 100.0 | 0.0079 |
| 300.0 | 0.0080 |
| 1000.0 | 0.0081 |
| 3000.0 | 0.0080 |
| 10000.0 | 0.0082 |
| 30000.0 | 0.0082 |

(b) SDF constraint medium level.

Table 9: SDF constraint ablation studies.

| Max delta fraction | Medium level error | Fine level error |
|---|---|---|
| 1.01 | 0.0098 | 0.0048 |
| 1.05 | 0.0098 | 0.0048 |
| 1.10 | 0.0100 | 0.0049 |
| 1.20 | 0.0101 | 0.0049 |
| 1.30 | 0.0103 | 0.0050 |
| 1.50 | 0.0106 | 0.0051 |
| 2.00 | 0.0113 | 0.0053 |
| 5.00 | 0.0139 | 0.0066 |

Table 10: Ablation studies of the delta factor. We multiply the delta by the values in the first column and measure the SDF error compared to the Open3D calculated SDF, which we use an ground-truth.

