# OpenReview forum: "Smooth Real-time Rendering via Implicit Nested Neighborhoods"
_ICLR.cc/2025/Conference — Submitted to ICLR 2025_

### Official Review · Reviewer_WGFb · 2024-10-28

**Soundness:** 2
**Presentation:** 2
**Contribution:** 2
**Rating:** 3
**Confidence:** 4

**Summary:**

The paper proposes a method for representing and rendering 3D shapes using implicit neural representations (INRs). Starting from an oriented and colored point cloud, it fits an SDF network with color and normal information. The network operates at three levels of detail to accelerate sphere tracing: a coarse, fast network for regions far from the surface, and more precise, slower networks when closer to the surface. An orthogonal projection maps surface normals and colors from detailed to coarser representations, preserving visual details, and accelerating rendering. Additionally, the paper introduces a GEMM (general matrix multiplication)-based approach for computing surface normals, which is twice as fast as PyTorch autograd.

**Strengths:**

* Introduces a multi-level network approach to accelerate sphere tracing.
* Employs orthogonal projection to transfer visual details to coarser representations.
* Proposes a GEMM-based method for faster computation of surface normals.
* Overall, many aspects of surface implicit rendering are discussed

**Weaknesses:**

* The purpose and practical use cases of the pipeline are not clearly defined:
    * If rendering speed is the main purpose, how does it compare to simply rendering a mesh? Meshes are discarded very early because they are “not efficient, nor continuous, nor scalable” (line 40). But the presented approach requires roughly 150 seconds to process a moderately complex input geometry (Armadillo, table 1), on an RTX 3090, so scalability is also a concern. At least, there is a tradeoff between rendering speed, memory consumption, and pre-processing. That should be mentioned and quantified.
    * One motivation is to “fully use the smoothness of INRs”, but this is never quantified or motivated. When do we need shapes to be very smooth? Ironically, the “lack of sharp features” is cited as a limitation of the method. Therefore I struggle to get the motivation about smoothness.
    * Demonstrations are limited to a few synthetic examples. Applicability to real, noisy RGB point clouds is uncertain.


* Some contributions, like the acceleration of marching cubes (Sec 4.5), are already established in existing literature - see BACON (Lindell et al.)
* One important technical mistake: In sec. 4.5, the interpolation between genus 0 and genus 1 shapes is presented as a “demonstration that [the] representation can be integrated into differentiable pipelines”. But this is pure interpolation, not driven by any gradient descent.


* Clarity can be improved:
    * About the GEMM-based acceleration of normal computations:
        * Section 3.6 is not motivating it. Why is it required for the paper’s objective of fast rendering? How much would the frame rate drop?
        * It also fails to convey an intuition as to why the method is faster than pytorch: less FLOP? Fused operation?
    * The abstract and introduction uses confusing terminology without proper upfront definitions. For example, the abstract mentions “localised SDF training based on nested neighborhoods”, but the reader cannot  know what it means before having read the paper. So this is not descriptive, whereas a “network ensemble with 3 levels of details” would be clearer.

* Minor issues:
    * delayed acronym definitions of GEMM (used on line 53, defined on line 80)
    * misplacement of tables and figures disrupt reading flow: for example Tab. 5 is mentioned on page 7, but appears on page 10.

Overall, the paper reads like a collection of techniques that are not really novel or insightful, and are sometimes poorly explained or unmotivated. The aggregate of these techniques does not fulfil a clear cut objective.

**Questions:**

Based on the above weaknesses:
* What are the intended purposes and practical scenarios for this pipeline? Can the method work on real, noisy RGB point clouds, or is it limited to synthetic data?
* How does this method compare to traditional mesh rendering techniques in terms of efficiency and scalability?
* When is the smoothness of INRs necessary?
* Why is the GEMM approach faster than PyTorch autograd? Does it reduce computations, use fused operations?
* What specific problem does the GEMM-based acceleration address? How much does the frame rate drop without it?

---

> ### Author Response · Authors · 2024-11-23
>
> We thank the reviewer for the detailed comments. Next, we address the raised questions.
>
> > **Contextualization with triangle meshes and clarifications about smoothness and practical scenarios of our pipeline.**
>
> Our objective is not to substitute triangle meshes, since we show that using our multiscale SDF representation allows fast mesh extraction using marching cubes (Sec. 4.3). Rather, we aim to provide an implicit representation for SDFs suitable for specific tasks depending on two main properties: **(1) high order differentiability (smoothness)** and **(2) fast rendering** (especially when surface extraction is otherwise prohibitive).
>
> Section 4.3 provides an application on neural surface evolution that explores those properties. Next we provide more details of this application, its association with the above properties, and why meshes are not suited for that case.
>
> **Smoothness**. Neural implicit surface evolution (NISE) [1] is a method that trains dynamic SDFs (surface evolution through time) using regularizations based on the level-set method. In addition to the Eikonal regularization necessary for the SDF, those losses need high-order derivatives to calculate differential properties of the surface (e.g. mean curvature). Since our SDF representation is smooth, those derivatives are available, making its integration with such a pipeline easy. Conversely, using meshes for surface evolution is challenging because the representation should be adapted to handle the lack of differentiability. Finally, meshes cannot easily handle topology changes (such as the hole created in the Bob model shown in Figure 11). Creating holes in the mesh during animation is a hard task due to its fixed topology. This problem is easily avoided using our implicit representation.
>
> **Fast rendering**. For rendering the resulting animation, the zero-level sets must be evaluated fast along time for the process to be real-time. This is naturally achieved by integrating our method with the NISE pipeline. On the other hand, doing the same rendering using meshes is prohibitive because the mesh should be extracted for each instant of the animation, which cannot be done in real-time because mesh extraction is not real-time. Preprocessing the animation is also prohibitive because each mesh extraction may take dozens of megabytes, creating an unacceptable memory footprint.
>
> We added in the Appendix A.1 a detailed discussion about the importance of the smoothness and real-time rendering properties of our representation.
>
> > **Acceleration of marching cubes (Sec 4.5), are already established in existing literature.**
>
> We apologize if we could not communicate correctly, but our intent is not to claim to be the first method to propose such an acceleration, but to describe additional applications of our representation to emphasize its flexibility. In addition to the fast sphere-tracing-based rendering, a surface may also be extracted using the accelerated marching cubes algorithm. Our objective with this application is to settle the discussion about the use of implicit or explicit representations. The user may transit to an explicit mesh representation in case they need or they may keep using the implicit SDF in case extracting a mesh is prohibitive or smoothness is needed.
>
> > **Clarification on the Interpolation between genus 0 and genus 1 shapes and why it is driven gradient descent.**
>
> NISE [1] does not apply simple interpolation between the surfaces, but trains an implicit neural representation (a coordinate network with an additional time parameter). Thus, the gradient descent is part of the approach. More specifically, the process is guided by a differential equation enforced by a regularization term in the loss. In addition to the Eikonal term, which ensures that the function is an SDF for each time instant, the interpolation term uses the Level-set equation to control the evolution of the surface. A boundary condition is associated with the SDF of the surface being interpolated, forcing the function to be equal to those SDFs in specific simulation times. The application we show in our paper is to perform the simulation in real-time by integrating our multiscale representation with the NISE pipeline. This is only possible because our representation is smooth, i.e. the derivatives needed for the regularization terms in the loss function of NISE’s surface interpolation are properly defined. That application would be much more difficult to achieve if the surfaces were triangle meshes, as explained in the point above.

---

> > ### Author Response · Authors · 2024-11-23
> >
> > > **Clarification on the importance of real-time rendering in our context.**
> >
> > Our objective with providing real-time rendering for our representation is to make its integration in neural pipelines more appealing. Fast rendering of neural SDFs may be useful for approaches that perform inverse rendering in that context. Training could be more efficient if the surface rendering is faster. Previous works that propose such pipelines include DIST [2] and SDFDiff [3]. SDFs are also popular to adapt training of volumetric representations to surfaces (NeuS [4]). We believe that tools to deal with SDFs in real-time may be beneficial for the community and our approach tries to fill this gap.
> >
> > > **Intuition about why our GEMM-based normal computation is faster than pytorch.**
> >
> > We believe the fact that our approach is analytic, depending only on forward passes of the network instead of backward passes as in pytorch, makes it faster. Our algorithm does not need to maintain a computational graph nor to traverse it, differently from pytorch. However, quantitatively showing that is difficult because that would require a profiling of inner parts of pytorch, which we believe is outside the scope of the work.
> >
> > > **Do we use fully fused operations in the GEMM-based normal computation?**
> >
> > No, we do not use fully fused operations. This is future work and we agree that it has potential to improve performance.
> >
> > > **What specific problem does the GEMM-based acceleration address? How much does the frame rate drop without it?**
> >
> > The GEMM-based acceleration addresses the problem of calculating surface normals for lighting calculation fast, without the need of the pytorch library. Since our normal computation is only based on forward passes of the MLP, our renderer may be fully implemented with a GEMM library (CUTLASS, for example) for inference and CUDA kernels for rendering. This is the current implementation used in the experiments of the paper. Pytorch is only needed for training.
> >
> > > **Can the method work on real, noisy RGB point clouds, or is it limited to synthetic data?**
> >
> > We are working on additional experiments, which will be added in this thread as we finish.
> >
> > **References:**
> >
> > [1] Novello, Tiago, et al. "Neural Implicit Surface Evolution." Proceedings of the IEEE/CVF International Conference on Computer Vision. 2023.
> >
> > [2] Liu, Shaohui, et al. "Dist: Rendering deep implicit signed distance function with differentiable sphere tracing." Proceedings of the IEEE/CVF Conference on Computer Vision and Pattern Recognition. 2020.
> >
> > [3] Jiang, Yue, et al. "Sdfdiff: Differentiable rendering of signed distance fields for 3d shape optimization." Proceedings of the IEEE/CVF conference on computer vision and pattern recognition. 2020.
> >
> > [4] Wang, Peng, et al. "NeuS: Learning Neural Implicit Surfaces by Volume Rendering for Multi-view Reconstruction." Advances in Neural Information Processing Systems.

---

> ### Author Response · Authors · 2024-11-27
> **New experiments available!**
>
> **We finished our additional experiments!**
>
> They are related with your questions and we would be very thankful if you could consider them in a reevaluation of the paper. They are on the Appendix of the new paper version. Specifically:
>
> 1. We updated Figure 13 to include a case with noisy input data. We used GS-LRM [1] to extract a noisy point cloud from images and successfully trained our multiscale surface representation.
>
> 2. We integrated our multiscale representation with NeuS [2], since it uses SDFs to accurately reconstruct 3D surfaces. Our representation improved NeuS, making it able to learn a high-fidelity surface in less iterations.
>
> 3. We trained our representation on a more detailed surface (Asian Dragon) and compared it against NGLOD [3] and IDF [4]. Our approach was able to present fine details, maintaining real-time performance.
>
> *We thank the reviewer for the patience.*

---

> > ### Author Response · Authors · 2024-11-27
> >
> > **References**
> >
> > [1] Zhang, Kai, et al. "Gs-lrm: Large reconstruction model for 3d gaussian splatting." European Conference on Computer Vision. Springer, Cham, 2025.
> >
> > [2] Wang, Peng, et al. "NeuS: Learning Neural Implicit Surfaces by Volume Rendering for Multi-view Reconstruction." Advances in Neural Information Processing Systems.
> >
> > [3] Takikawa, Towaki, et al. "Neural geometric level of detail: Real-time rendering with implicit 3d shapes." Proceedings of the IEEE/CVF Conference on Computer Vision and Pattern Recognition. 2021.
> >
> > [4] Yifan, Wang, Lukas Rahmann, and Olga Sorkine-hornung. "Geometry-Consistent Neural Shape Representation with Implicit Displacement Fields." International Conference on Learning Representations.

---

### Official Review · Reviewer_zpqm · 2024-10-30

**Soundness:** 3
**Presentation:** 3
**Contribution:** 3
**Rating:** 6
**Confidence:** 4

**Summary:**

The paper proposes to accelerate sphere tracing using hierarchical SDFs achieving real time performance. The neural SDFs are trained based on nested neighborhood efficiently. The paper also proposes a GEMM-based normal computation. The experiments show that textured 3D models are able to be rendered in real time without surface extraction.

**Strengths:**

The work generalizes sphere tracing method to multiscale sphere tracing. Taking advantage of the multiscale SDFs, the method is able to accelerate the sphere tracing.

The method is also able to render highly fidelity textures. The results look good in the experiments.

The method is highly efficient. It accelerates the rendering efficiency several times than SOTA.

**Weaknesses:**

To more neural SDFs increases the training time and storage space. It is also slow for mesh extraction. Moreover, the residual SDFs are not stable to learn, which may result in artifacts.

The parameters $\theta_i$ may be sensitive. Smaller ones may result in errors, but larger ones affect the efficiency. How to trade off them? Is it appropriate to use global $\theta_i$? Perhaps, location varying $\theta_i$ is a better choice.

**Questions:**

I wonder the geometric accuracy of the rendering surface. This is an important factor to evaluate the method.

IDF took about 40 mins for training, but it took only 100s in this work. The proposed method took about 150s for training, which is also efficient. How does the work achieve such improvement?

The rendering results are glossy. Does the method support BRDF material for rendering.

---

> ### Author Response · Authors · 2024-11-23
>
> We thank the reviewer for the evaluation of our work and comments. Next we address the raised questions.
>
> > **To more neural SDFs increases the training time and storage space. It is also slow for mesh extraction. Moreover, the residual SDFs are not stable to learn, which may result in artifacts.**
>
> It is not necessarily true that more neural SDFs increases training time and storage space. Once the base SDF is trained over the entire domain, supervision for the residuals is needed only in a small neighborhood of the original surface. This localized supervision allows the residual INRs to be trained faster. Additionally, we observed that fewer parameters are sufficient to represent the same surface. This is because supervising residuals in regions near the data (a 2D subset of the space) is more efficient for encoding fine details.
> To illustrate this, we compared our multiscale SDF with a single MLP having 3 hidden layers of 256 neurons, using the Armadillo model. Training the single MLP took 391 seconds but failed to achieve the same level of detail as our method. In contrast, our multiscale SDF, configured as shown in Table 1—MLPs with sizes (64, 1), (128, 1), and (256, 1)—achieved better detail with a total training time of only 148.9 seconds.
>
> Regarding storage space, adding the disk-space of the base and residual MLPs used 355 KB (20 KB (coarse) + 70 KB (medium) + 265 KB (fine)), while it required 780 KB for the single MLP case. Thus, our multiscale SDF offers a more compact representation.
>
> Regarding slow mesh extraction, our representation enables faster extraction by leveraging the multiscale SDF. This is shown in Table 4, where "no culling" refers to standard marching cubes, while "culling" represents our proposed adaptation, which achieves a between 4x and 7x performance improvement.
>
> Finally, regarding the stability of the residual SDF, Figure 10 shows that using residuals mitigates spurious artifacts in reconstruction. This is evident when comparing the top row (without residuals) to the bottom row (with residuals, ours).
>
> > **The parameters θi may be sensitive. Smaller ones may result in errors, but larger ones affect the efficiency. How to trade off them? Is it appropriate to use global θi? Perhaps, location varying θi is a better choice.**
>
> We assume that by θi the reviewer is referring to the delta parameter. Indeed, defining larger deltas affects the efficiency of sphere tracing as shown in Figure 4. However, to avoid manually defining, Sec. 3.3 introduces a procedure to compute them directly from the data while ensuring that the nesting condition (Eq (1)) holds – crucial for the rendering. We found that this method yields a good trade-off between efficiency and quality.
> Finally, we think defining local deltas (depending on the position) would increase computation time since during rendering we would need to add their inference. It is also not clear to us how to adapt Sec 3.3 to this setting.
>
> > **I wonder the geometric accuracy of the rendering surface. This is an important factor to evaluate the method.**
>
> Table 2 (in the paper) presents the Hausdorff distance between the zero-level set of the trained SDF and the ground-truth surface. All values are on the order of $10^{-4}$, demonstrating the high accuracy of the approximation.
>
> > **IDF took about 40 mins for training, but it took only 100s in this work. The proposed method took about 150s for training, which is also efficient. How does the work achieve such improvement?**
>
> We’ve followed the instructions in IDF’s README, employing the same architecture and hyperparameters. However, they employed a total of *8 million samples*, half on-surface, and half off-surface. The largest surface we used has ~543k vertices, thus 8 million samples would be overkill. For Tab. 1, we used the Armadillo, which has 172k vertices. For IDF training, we used a total of 344k samples (172k * 2), which might explain the difference in training times, even though the network architecture follows the one proposed in IDF.
>
> > **The rendering results are glossy. Does the method support BRDF material for rendering.**
>
> Our method supports BRDFs. The marching cube cases in Figs. 8 and 9 were rendered in real-time using NVIDIA's Omniverse’s ray-tracing pipeline with BRDFs. The normal and texture mapping is done using our approach.

---

> ### Comment · Reviewer_zpqm · 2024-11-26
>
> Thank you for the reply. I choose to keep my score.

---

> ### Author Response · Authors · 2024-11-27
> **New experiments available!**
>
> **We thank the reviewer for his time and patience.**
>
> In case you would like to see, we conducted promising new experiments that are available in the appendix of the new version of the paper. Specifically:
>
> 1. We updated Figure 13 to include a case with noisy input data. We used GS-LRM [1] to extract a noisy point cloud from images and successfully trained our multiscale surface representation.
>
> 2. We integrated our multiscale representation with NeuS [2], since it uses SDFs to accurately reconstruct 3D surfaces. Our representation improved NeuS, making it able to learn a high-fidelity surface in less iterations.
>
> 3. We trained our representation on a more detailed surface (Asian Dragon) and compared it against NGLOD [3] and IDF [4]. Our approach was able to present fine details, maintaining real-time performance.

---

> > ### Author Response · Authors · 2024-11-27
> >
> > **References**
> >
> > [1] Zhang, Kai, et al. "Gs-lrm: Large reconstruction model for 3d gaussian splatting." European Conference on Computer Vision. Springer, Cham, 2025.
> >
> > [2] Wang, Peng, et al. "NeuS: Learning Neural Implicit Surfaces by Volume Rendering for Multi-view Reconstruction." Advances in Neural Information Processing Systems.
> >
> > [3] Takikawa, Towaki, et al. "Neural geometric level of detail: Real-time rendering with implicit 3d shapes." Proceedings of the IEEE/CVF Conference on Computer Vision and Pattern Recognition. 2021.
> >
> > [4] Yifan, Wang, Lukas Rahmann, and Olga Sorkine-hornung. "Geometry-Consistent Neural Shape Representation with Implicit Displacement Fields." International Conference on Learning Representations.

---

### Official Review · Reviewer_ix5A · 2024-10-31

**Soundness:** 3
**Presentation:** 2
**Contribution:** 3
**Rating:** 6
**Confidence:** 4

**Summary:**

The paper presents some novel ideas for implicit 3D representation and rendering: nested implicit surfaces for multi-level surface details, neural attribute mapping for highly detailed normals and colors, and a corresponding multiscale sphere tracing algorithm to render these surfaces in realtime. The authors demonstrate a couple of useful applications of their representation for surface morphing and speeding up marching cubes.

**Strengths:**

The nested implicit functions, where next level is within a small shell of the previous level, is a natural and convenient way to achieve multiple scales of detail. Mapping attributes from an even higher-detail implicit than the surface is a good way to increase detail while keeping the sphere-tracing time shorter.

**Weaknesses:**

The writing could be improved to read more smoothly. I will outline major feedback here and more detailed comments under Questions.

The abstract does not clearly present the gist of this work. The first paragraph seems to serve only as related work overview, but is missing motivation. 1) Why do we care about surface representation -- the proposed representation can encode enough visual detail for rendering in games/movies and can easily be integrated into ever-improving ML algorithms? 2) Why do we care about realtime rendering -- the proposed algorithm can be used to speed up training (when minimizing an error between a rendering and a target image), and/or is somehow more suitable for interactive applications (games) than triangles? The second paragraph is clear until the very end where some applications are listed  -- does this representation unlock new applications, or is it just applicable in these applications along with other representations? You may want to just say "we demonstrate the utility of our representation in several real-world applications."

The paper mentions "normals and textures" several times. This should probably be "normals and colors" as those are the attributes it learns to represent and compute at each point. Texture invokes traditional "texture mapping" where a coarse surface representation can use texture coordinates to interpolate much higher-resolution (and not necessarily smooth) colors from an image.  I think the paper would be clearer if it says color instead of texture.

It seems that effort has been put into squeezing as much as possible into the 10 page limit, which results in a busy layout and no references to figures tables in the text. This complicates the flow of the paper, as we do not necessarily know when to refer to a certain figure.

Figure 2: How come q1 is not on the surface s1? If q2 is on S2 and q3 is on S3, it would feel more consistent if q1 was on S1. Is this because the view ray needs to be traced until the most detailed surface point is reached (which is on S2), and then a different normal ray can be traced to find the point on S3 that only contributes the attribute value (not position)? In that case, N2 is unnecessary and only clutters the image.  In addition, looking at algorithm 1, even q2 should not lie on S2, because for j < 3 we trace until Sj-dj. Maybe more work is needed to reconcile the figure and the algorithm.

Finally, I believe a more suitable venue for this would would be a vision or graphics conference (e.g. CVPR, ICCV, Siggraph).

**Questions:**

Line 049: "GEMM-based normal computation" should be accompanied by the GEMM reference, as it is the first mention of it.

First line of Overview should spell out "Sphere Tracing (ST)" as its initial definition is deep in the dense previous work paragraphs.

Figure 1: By "normal mapping algorithms" did you mean to say a more general "attribute mapping algorithms"?

Line 153: Phrase "compute it first intersection point q2" should be fixed to "compute its S2 intersection point q2".

Figure 3: Should "using too large d1 == d2" instead say "using too large d1 or d2"? Does it matter that they are equal, or does it matter that they are too large?

Reference to "Local Deep Implicit Functions for 3D Shape" by Genova et al. could be added as an example of using multiple smaller implicit representations to increase detail of the whole representation.

Section 4, why don't you just let (N, d) mean MLP with "d" hidden layers? Why the need to add 1, it only adds confusion.

Figure 7: Could you add shaded surfaces, so that we can judge how good the normals are? It is hard to judge the normal visualization.

Figure 9 is quite confusing as it covers multiple concepts. What does the left-side vs right-side of the middle image (armadillo) mean? Is one a mesh and the other a mesh with neural normal mapping?

---

> ### Author Response · Authors · 2024-11-23
>
> We thank the reviewer for the kind evaluation and suggestions.
>
> > **Fixes and reference to “Local Deep Implicit Functions for 3D Shape” by Genova et al.**
>
> We adopted all the small fixes, reflected in the new uploaded version.
>
> > **Why do we care about surface representation?**
>
> The components of our representation have potential to be used in general 3D objects in games / movies. This is shown in Figures 8 (except the first case) and 9 (middle), which are coarse triangle meshes extracted via marching cubes and rendered in real-time using NVIDIA’s Omniverse ray tracer, in conjunction with our neural texture and normal mapping algorithms. This mapping is done without any additional parameterization. We also propose a marching cubes optimization in Section 4.3 for better transit between our representation and triangle meshes.
>
> However, the best potential of our multiscale SDF representation is in integration with other INR pipelines. Beyond its standard application in 3D reconstruction from point clouds, we show a concrete integration case with Neural implicit surface evolution (NISE) [1] in Section 4.3. This is enabled due to two key properties of our multiscale SDF: (1) **smoothness**, which helps defining the loss functions using differential equations, and (2) **fast rendering**, crucial for visualizing the surface evolution – in this case mesh extraction is unfeasible.
>
> **Smoothness.** NISE [1] is a method that trains dynamic SDFs (surface evolution through time) using regularizations based on the level-set method. In addition to the Eikonal regularization necessary for the SDF, those losses need high-order derivatives to calculate differential properties of the surface (e.g. mean curvature). Since our SDF representation is smooth, those derivatives are available, making its integration with such a pipeline straightforward. Conversely, using meshes for surface evolution is challenging because the representation should be adapted to handle the lack of differentiability. Finally, meshes cannot easily handle topology changes (such as the hole created in the Bob model shown in Figure 11). Creating holes in the mesh during animation is a hard task due to its fixed topology. This problem is easily avoided using our implicit representation.
>
> **Fast rendering.** For rendering the resulting animation, the zero-level sets must be evaluated quickly along time for the process to be real-time. This is naturally achieved by integrating our method with the NISE pipeline. On the other hand, doing the same rendering using meshes is prohibitive because the mesh should be extracted for each instant of the animation, which cannot be done in real-time. Preprocessing the animation is also prohibitive because each mesh extraction is costly in terms of memory and processing power.
>
> We added in the Appendix A.1 a detailed discussion about the importance of the smoothness and real-time rendering properties of our representation.
>
> > **Clarification on the importance of real-time rendering in our context**
>
> Our objective with providing real-time tools for our multiscale SDF is to enhance its appeal for integration into neural pipelines. Fast rendering of neural SDFs is valuable for approaches involving inverse rendering, where fast surface rendering is important for efficient training. Previous works that propose such pipelines include DIST [2] and SDFDiff [3]. SDFs are also popular in surface reconstruction from images (eg. NeuS [4]). We believe that real-time tools for handling SDFs could greatly benefit the community, and our approach seeks to address this gap.
>
> > **Clarification on the use of the term  "texture mapping".**
>
> The reason behind the use of the term “texture mapping” is to describe the process as a mapping of the texture of a surface into another surface. Traditional texture mapping indeed fetches colors from an image at rendering time, however this is done after the texture image is created (baked) from a surface in a pre-processing step. The mapping is then done via UV-parameterization. Our approach makes a mapping between the surfaces without the need of the additional parameterization, which is a good property for a surface representation. Another reason is that texture mapping is widely known and we believe our approach is easily understood when associated with such a familiar technique.
>
> > **About suitability for the conference.**
>
> Part of our contributions is to propose a learned multi-level surface representation, which encompasses attributes such as colors, normals and texture, which we believe fits the conference main theme of learning representations. Additionally, prior work on surface representations were present at the conference, such as “Geometry-Consistent Neural Shape Representation with Implicit Displacement Fields” [5], published at ICLR-2022.

---

> ### Author Response · Authors · 2024-11-23
>
> > **Section 4, why don't you just let (N, d) mean MLP with "d" hidden layers? Why the need to add 1, it only adds confusion.**
>
> The reviewer is correct. This is actually a typo;  (N, d) means a MLP with “d” hidden layers of the form RN→ RN.
>
> > **Figure 9 is quite confusing as it covers multiple concepts. What does the left-side vs right-side of the middle image (armadillo) mean? Is one a mesh and the other a mesh with neural normal mapping?**
>
> Thank you for pointing this out; we have improved the figure. Regarding the middle figure, the reviewer is correct. Its left-side presents the mesh extracted from the coarse SDF of the Armadillo and its right-side gives the same mesh redented using the proposed normal mapping.
>
> > **Figure 3: Should "using too large d1 == d2" instead say "using too large d1 or d2"? Does it matter that they are equal, or does it matter that they are too large?**
>
> The reviewer is correct, but we maintained the equality because we performed the experiment with equal deltas.
>
> **References:**
>
> [1] Novello, Tiago, et al. "Neural Implicit Surface Evolution." Proceedings of the IEEE/CVF International Conference on Computer Vision. 2023.
>
> [2] Liu, Shaohui, et al. "Dist: Rendering deep implicit signed distance function with differentiable sphere tracing." Proceedings of the IEEE/CVF Conference on Computer Vision and Pattern Recognition. 2020.
>
> [3] Jiang, Yue, et al. "Sdfdiff: Differentiable rendering of signed distance fields for 3d shape optimization." Proceedings of the IEEE/CVF conference on computer vision and pattern recognition. 2020.
>
> [4] Wang, Peng, et al. "NeuS: Learning Neural Implicit Surfaces by Volume Rendering for Multi-view Reconstruction." Advances in Neural Information Processing Systems.
>
> [5] Yifan, Wang, Lukas Rahmann, and Olga Sorkine-hornung. "Geometry-Consistent Neural Shape Representation with Implicit Displacement Fields." International Conference on Learning Representations.

---

> > ### Author Response · Authors · 2024-11-27
> > **New experiments available!**
> >
> > **We finished our additional experiments**!
> >
> > They are on the Appendix of the new paper version. Specifically:
> >
> > 1. We updated Figure 13 to include a case with noisy input data. We used GS-LRM [1] to extract a noisy point cloud from images and successfully trained our multiscale surface representation.
> >
> > 2. We integrated our multiscale representation with NeuS [2], since it uses SDFs to accurately reconstruct 3D surfaces. Our representation improved NeuS, making it able to learn a high-fidelity surface in less iterations.
> >
> > 3. We trained our representation on a more detailed surface (Asian Dragon) and compared it against NGLOD [3] and IDF [4]. Our approach was able to present fine details, maintaining real-time performance.
> >
> > *We thank the reviewer for the patience and politely ask for consideration of those results and reevaluation of the submission.*

---

> > > ### Author Response · Authors · 2024-11-27
> > >
> > > **References**
> > >
> > > [1] Zhang, Kai, et al. "Gs-lrm: Large reconstruction model for 3d gaussian splatting." European Conference on Computer Vision. Springer, Cham, 2025.
> > >
> > > [2] Wang, Peng, et al. "NeuS: Learning Neural Implicit Surfaces by Volume Rendering for Multi-view Reconstruction." Advances in Neural Information Processing Systems.
> > >
> > > [3] Takikawa, Towaki, et al. "Neural geometric level of detail: Real-time rendering with implicit 3d shapes." Proceedings of the IEEE/CVF Conference on Computer Vision and Pattern Recognition. 2021.
> > >
> > > [4] Yifan, Wang, Lukas Rahmann, and Olga Sorkine-hornung. "Geometry-Consistent Neural Shape Representation with Implicit Displacement Fields." International Conference on Learning Representations.

---

### Official Review · Reviewer_FrXv · 2024-11-05

**Soundness:** 3
**Presentation:** 3
**Contribution:** 3
**Rating:** 6
**Confidence:** 3

**Summary:**

The paper presents a method for learning implicit neural representation of smooth closed surfaces (like small objects) with texture and normal, initialized with point cloud and normals. The proposed method is end to end differentiable allowing direct use of the INR for rendering, avoiding the need to extract explicit triangle meshes.

This is achieved by learning the surface representation as multiscale SDF and rendering with multi-scale sphere tracing. The multiscale SDFs are defined in a nested manner, coarse surface S1 - coarsest SDF f1, S2 is nested in the neighborhood of f1, and so on. The nested SDFs are rendered using proposed multiscale sphere tracing. The authors also show analytical computation of normals using GEMM which is fast (does not need auto-grad).

The results are shown on classic toy dataset, single objects like Armadilo, Buddha, Bunny, etc. and compared with two methods IDF and NGLOD.

**Strengths:**

Originality and quality: The paper proposes a sound multiscale SDF representation and show how such a representation can be effectively learned and used for real-time rendering. The paper builds on top of concepts from previous works, but to the best of my knowledge, the specific proposed representation and corresponding learning framework (loss formulation, nested neighborhoods, attribute mapping) are novel and well-grounded. Experiments provide decent validation but conducted on a small number of simple synthetic objects.

Clarity: The paper is generally written well and provides adequate level of detail with clear notation, however the explanation can be made clearer (more on this in Weaknesses).

Significance: While the results are evaluated on toy dataset with pristine inputs, the author show one example of real-world usecase in Fig 13. The limitation of smooth surfaces (cannot represent sharp edges) is somewhat limiting for real-world objects but in my opinion the community will find value in the proposed approach and such limitations can perhaps be overcome by future works.

**Weaknesses:**

- The paper is decently written but the authors can improve the writing and flow of information better to make it easier to follow. I had to go back and forth a few times to get to the key ideas. Section 3.1 Overview can be used to provide gist of the framework since notations are subsequently introduced in good detail. Section 3.5 can also be explained more clearly.

- It is mentioned in the abstract: "Extracting meshes is not a real-time task and introduces unnecessary discretization to rendering". There are many advantages of explicit triangle meshes, while extracting meshes is not a real-time task, learning SDFs is also not a real-time task. I don't find this motivation fully convincing and I would be curious to know how the method fares with respect to explicit meshing from SDF in terms of rendering quality and performance.

- A relevant work is Deep Marching Tetrahedra (Shen et al) that learns implicit surface representation and extracts a mesh in a differentiable manner. There are also follow up works. I think this class of methods are worth discussing and comparing against.

**Questions:**

- Did you experiment with more complex real-world datasets such as ShapeNet or 3D objects from Objaverse?
- Given the end-to-end differentiable nature of the proposed approach, will this method be applicable for image based surface representation learning? [similar to VolSDF (Yariv et al) and follow ups]

---

> ### Author Response · Authors · 2024-11-23
>
> We thank the reviewer for the encouraging comments. Next, we clarify the raised points.
>
> > **Extracting meshes is not a real-time task but learning SDFs is also not a real-time task.**
>
> Indeed. However, our comment was referring to the specific task of visualizing the zero-level set of a trained neural SDF. We added a section in the Appendix (A.1) that clarifies the contextualization with triangle meshes. Extracting a mesh from a trained neural SDF results in a substantial memory footprint, especially when the zero-level set is highly detailed. This is primarily due to the cubic complexity of grid generation for marching cubes. For example, we trained a multiscale SDF using the following architecture: $(128,1)$ for the coarse level, $(256,1)$ for the medium level, and $(256, 2)$ for the fine level. Generating the grid of resolution $512^3$  and running the marching cubes for that case demands approximately 20 GB of GPU memory while rendering with our sphere tracing using an image resolution of $512^2$ requires significantly less—approximately 5 GB, including the GEMM buffers used to parallelize the pixel computation. Additionally, storing high-resolution meshes is costly in terms of memory. For this experiment, the output mesh has 43 MB of storage, while the underlying multiscale MLP representation needs only 857 KB, showing that our representation is significantly more compact.
>
> > **Contextualization with triangle meshes**
>
> Our objective is not to substitute triangle meshes, in fact, we show that using our multiscale INR allows fast mesh extraction using marching cubes (Section 4.3).
> Rather, we aim to provide an implicit representation for SDFs suitable for specific tasks depending on two main properties:
>
> 1. high order differentiability (smoothness)
> 2. fast rendering (especially when surface extraction may be prohibitive).
>
> Section 4.3 gives an application on neural surface evolution that explores those properties.
>
> **Smoothness**. Neural implicit surface evolution (NISE) [1] is a method that trains dynamic SDFs (surface evolution through time) using the level-set method. Thus, in addition to the Eikonal regularization necessary for the SDF, we need high-order derivatives to compute differential properties (e.g. mean curvature). Since our INR is smooth, those derivatives are available, making its integration with such a pipeline easier. Conversely, using meshes for surface evolution is challenging because the representation should be adapted to handle the lack of differentiability. Finally, meshes cannot easily handle topology changes (eg. the Spot-Bob interpolation shown in Figure 11). Creating holes in the mesh during animation is a hard task due to its fixed topology. This problem is easily avoided by our implicit representation.
>
> **Fast rendering**. For rendering the resulting animation, the zero-level sets must be evaluated fast during evolution for real-time visualization. This is achieved by integrating our method with the NISE pipeline. On the other hand, doing the same rendering using meshes is prohibitive because the mesh should be extracted for each instant of the animation, which cannot be done in real-time due to mesh extraction. Preprocessing the animation is also prohibitive because each mesh extraction may take dozens of megabytes (see the above experiment), creating an unacceptable memory footprint.
>
>
> Finally, an additional objective of providing real-time rendering for neural SDFs is making its integration in neural pipelines appealing. For example, fast rendering of such INRs is useful in inverse rendering tasks since it helps accelerate training. Previous works that propose such pipelines include DIST [2] and SDFDiff [3]. Additionally, SDFs are becoming a popular surface representation in 3D reconstruction from images using differentiable volume rendering (NeuS [4]). We believe that tools to deal with SDFs in real-time may be beneficial for the community and our approach tries to fill this gap.
>
> > **More complex datasets**
>
> We are conducting experiments on more complex surfaces and will share the results in this comment thread as soon as they are available.
>
> > **Clarification on image-based representation learning.**
>
> Figure 13 in the Appendix has an application that uses Depth Anything [5] to compute the depth of the image and create a point cloud from it. We then use the point cloud as input to our pipeline. This application shows the potential of the representation to learn surfaces from images.
>
> > **Writing improvements.**
>
> For any writing issues are present, we are already conducting a thorough proofreading and optimizing the structure and descriptions throughout the paper. We uploaded a new paper version with several changes.

---

> ### Author Response · Authors · 2024-11-23
>
> **References**
>
> [1] Novello, Tiago, et al. "Neural Implicit Surface Evolution." Proceedings of the IEEE/CVF International Conference on Computer Vision. 2023.
>
> [2] Liu, Shaohui, et al. "Dist: Rendering deep implicit signed distance function with differentiable sphere tracing." Proceedings of the IEEE/CVF Conference on Computer Vision and Pattern Recognition. 2020.
>
> [3] Jiang, Yue, et al. "Sdfdiff: Differentiable rendering of signed distance fields for 3d shape optimization." Proceedings of the IEEE/CVF conference on computer vision and pattern recognition. 2020.
>
> [4] Wang, Peng, et al. "NeuS: Learning Neural Implicit Surfaces by Volume Rendering for Multi-view Reconstruction." Advances in Neural Information Processing Systems.
>
> [5] Yang, Lihe, et al. "Depth anything: Unleashing the power of large-scale unlabeled data." Proceedings of the IEEE/CVF Conference on Computer Vision and Pattern Recognition. 2024.

---

> ### Author Response · Authors · 2024-11-26
> **New experiments available!**
>
> **We finished our additional experiments**!
>
> They are on the Appendix of the new paper version. Specifically:
>
> 1. We updated Figure 13 to include a case with noisy input data. We used GS-LRM [1] to extract a noisy point cloud from images and successfully trained our multiscale surface representation.
>
> 2. We integrated our multiscale representation with NeuS [2], since it uses SDFs to accurately reconstruct 3D surfaces. Our representation improved NeuS, making it able to learn a high-fidelity surface in less iterations.
>
> 3. We trained our representation on a more detailed surface (Asian Dragon) and compared it against NGLOD [3] and IDF [4]. Our approach was able to present fine details, maintaining real-time performance.
>
> *We thank the reviewer for the patience and politely ask for consideration of those results and reevaluation of the submission.*

---

> > ### Author Response · Authors · 2024-11-27
> >
> > **References**
> >
> > [1] Zhang, Kai, et al. "Gs-lrm: Large reconstruction model for 3d gaussian splatting." European Conference on Computer Vision. Springer, Cham, 2025.
> >
> > [2] Wang, Peng, et al. "NeuS: Learning Neural Implicit Surfaces by Volume Rendering for Multi-view Reconstruction." Advances in Neural Information Processing Systems.
> >
> > [3] Takikawa, Towaki, et al. "Neural geometric level of detail: Real-time rendering with implicit 3d shapes." Proceedings of the IEEE/CVF Conference on Computer Vision and Pattern Recognition. 2021.
> >
> > [4] Yifan, Wang, Lukas Rahmann, and Olga Sorkine-hornung. "Geometry-Consistent Neural Shape Representation with Implicit Displacement Fields." International Conference on Learning Representations.

---

### Meta-Review · Area_Chair_Nt4D · 2024-12-14

**Metareview:**

The paper addresses real-time rendering of closed surfaces modeled by neural implicit representations. While prior work trades off accuracy or differentiability for rendering speed, the paper proposes speeding up the rendering through sphere tracing of a multi-scale representation. A coarse base representation is smaller and can be evaluated quicker. Local nested representtions model fine details. To further speed up rendering, the paper proposes a novel approach to computing surface normals from the SDF.

The paper provides a detailed description of the approach. The proposed method is sound and claims are sufficiently validated on a small set of (toy) samples. The paper further demonstrates practicality through several examples of integrating the method into downstream applications.

The main problem of the paper is the writing. While the paper provides necessary details for all contributions and some validating experiments, it lacks a strong motivation. Reviewers questioned the need for real-time rendering of SDF-based approaches and asked for a comparison to mesh-based approaches in case rendering speed is of concern. The motivation was addressed in the discussion period through application for neural implicit surface evolution, but it remained an addition to the appendix. Despite being a major source of critique by reviewers, the writing and motivation of the paper was not substantially improved in the discussion period.

The paper could be substantially improved in a revision, which should clearly demonstrate the benefits of the proposed method.

**Additional Comments On Reviewer Discussion:**

- Reviewers [FrXv, ix5A, WGFb] criticized the writing. Authors promised proofreading, but the revision barely improved.
- unconvincing motivation [FrXv, WGFb, ix5A], authors argued for application to neural implicit surface evolution, added experiments integrating it into NeuS.
- limited novelty [WGFb]
- Validation is only provided on a small set of toy examples [FrXv]. The authors added another, larger sample.
- Limited applicability of representation through representation that only supports smooth surfaces [FrXv, WGFb]
- Missing discussion of related work [FrXv,WGFb], was added in revision.
- Unsuitable venue [ix5A], was disputed w/ given example of prior ICLR.
- requested comparison to mesh rendering [WGFb], was ignored.

---

### Decision · Program_Chairs · 2025-01-22

Reject